# Fine-tuning Language Models over Slow Networks using Activation Quantization with Guarantees

**Jue Wang**[1,3][*], **Binhang Yuan**[1][*], **Luka Rimanic**[1][†][*], **Yongjun He**[1], **Tri Dao**[2],
**Beidi Chen**[34], **Christopher Ré**[2], **Ce Zhang**[1]

[1]ETH Zürich, Switzerland   [2]Stanford University, USA   [3]Zhejiang University, China
[4]Carnegie Mellon University   [5]Meta AI

{juewang, binhang.yuan, luka.rimanic, yongjun.he, ce.zhang}@inf.ethz.ch
{beidic, trid, chrismre}@stanford.edu

## Abstract

Communication compression is a crucial technique for modern distributed learning systems to alleviate their communication bottlenecks over slower networks. Despite recent intensive studies of gradient compression for data parallel-style training, compressing the *activations* for models trained with pipeline parallelism is still an open problem. In this paper, we propose `AQ-SGD`, a novel activation compression algorithm for communication-efficient pipeline parallelism training over slow networks. Different from previous efforts in activation compression, instead of compressing activation values directly, `AQ-SGD` compresses the *changes of the activations*. This allows us to show, to the best of our knowledge for the first time, that one can still achieve $O(1/\sqrt{T})$ convergence rate for non-convex objectives under activation compression, without making assumptions on gradient unbiasedness that do not hold for deep learning models with non-linear activation functions. We then show that `AQ-SGD` can be optimized and implemented efficiently, without additional end-to-end runtime overhead. We evaluated `AQ-SGD` to fine-tune language models with up to 1.5 billion parameters, compressing activation to 2-4 bits. `AQ-SGD` provides up to $4.3\times$ end-to-end speed-up in slower networks, without sacrificing model quality. Moreover, we also show that `AQ-SGD` can be combined with state-of-the-art gradient compression algorithms to enable "end-to-end communication compression": *All communications between machines, including model gradients, forward activations, and backward gradients are compressed into lower precision*. This provides up to $4.9\times$ end-to-end speed-up, without sacrificing model quality.

## 1 Introduction

Decentralized or open collaborative training has recently attracted intensive interests [1, 2, 3, 4]. Despite their great potential in leveraging geo-distributed powerful GPUs, the computation efficiency is severely hindered by low network bandwidth — typically in the range of 10-400Mbps [2, 3, 4]. Recently, efforts in improving communication efficiency have significantly decreased the dependency on fast data center networks — the *gradient* can be compressed to lower precision or sparsified [5, 6, 7, 8], which speeds up training over low bandwidth networks, whereas the *communication topology* can be decentralized [9, 10, 11, 12, 13, 14], which speeds up training over high latency networks. Indeed, today's state-of-the-art training systems, such as Pytorch [15, 16], Horovod [17], Bagua [18], and BytePS [19], already support many of these communication-efficient training paradigms.

---

[*]Equal contribution.   [†]Now at Google.

36th Conference on Neural Information Processing Systems (NeurIPS 2022).

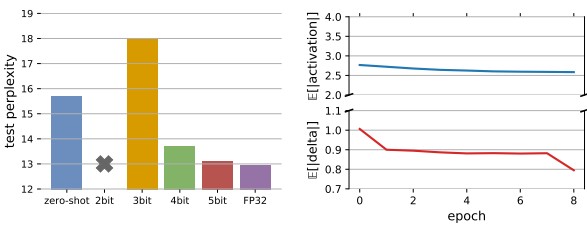

| (a) Fine-tune WikiText2 | (b) Activation and delta |

Figure 1: (a) Fine-tuning GPT2-1.5B with different activation precisions in communication; (b) Average absolute value of activations and their changes for GPT2-1.5B during training.

Table 1: Summary of technical results. AC-GC [29] and TinyScript [30] assume that the returned gradient is unbiased, whereas `AQ-SGD` algorithm does not rely on such an assumption.

| Algorithm | Assumptions on Quant. Grad. | Conv. Rate |
|---|---|---|
| SGD [31] | N/A | $\mathcal{O}(1/\sqrt{T})$ |
| AC-GC [29] | Unbiased | $\mathcal{O}(1/\sqrt{T})$ |
| TinyScript [30] | Unbiased | $\mathcal{O}(1/\sqrt{T})$ |
| AQ-SGD | N/A | $\mathcal{O}(1/\sqrt{T})$ |

However, with the rise of large foundation models [20] (e.g., BERT [21], GPT-3 [22], and CLIP[23]), improving communication efficiency via compression becomes more challenging. Current training systems for foundation models such as Megatron [24], Deepspeed [25], and Fairscale [26], allocate different layers of the model onto multiple devices and need to communicate — *in addition to* the gradients on the models — the *activations* during the forward pass and the *gradients on the activations* during the backward pass. Compressing these *activations* leads to a very different behavior compared with compressing the gradient — *simply compressing these activations in a stochastically unbiased way will lead to biases in the gradient that cannot be measured easily or expressed in closed form*. This either breaks the unbiasedness assumptions made by most gradient compression results [5, 6, 7, 8] or makes error compensation over gradient biases [27, 28] difficult to adopt.

Previous efforts on activation compression [32, 33, 34, 35, 36] illustrate, albeit mostly empirically, that large deep learning models can tolerate some compression errors on these activation values. However, when it comes to the underlying theoretical analysis, these efforts mostly make assumptions that do not apply to neural networks with non-linear activation functions — the only two recent efforts that claim theoretical convergence analysis [29, 30] assume that an unbiased compression on activations leads to an unbiased error on the gradient. Not surprisingly, these algorithms lead to suboptimal quality under relatively aggressive compression, illustrated in Figure 1a — in many cases, using activation compression to fine-tune a model might be worse than zero-shot learning without any fine-tuning at all.

In this paper, we focus on the problem of activation compression for training language models over slow networks by asking the following:

- **Q1.** *Can we design an algorithm for activation compression with rigorous theoretical guarantees on SGD convergence?*

- **Q2.** *Can such an algorithm be implemented efficiently without additional runtime overhead and outperform today's activation compression algorithms without accuracy loss?*

Our answers to both questions are *Yes*. **(Contribution 1)** We propose `AQ-SGD`, a novel algorithm for activation compression. The idea of `AQ-SGD` is simple — instead of directly compressing the activations, *compress the change of activations for the same training example across epochs*. Intuitively, we expect `AQ-SGD` to outperform simply compressing the activations because it enables an interesting "self-enforcing" dynamics: *the more training stabilizes → the smaller the changes of the model across epochs → the smaller the changes of activations for the same training example across epochs → the smaller the compression error using the same #bits → training stabilizes more.* **(Contribution 2)** The theoretical analysis of `AQ-SGD` is non-trivial since we have to analyze the above dynamics and connect it to SGD convergence, which is quite different from most of today's results on gradient compression and error compensation. Under mild technical conditions and quantization functions with bounded error, we show that `AQ-SGD` converges with a rate of $O(1/\sqrt{T})$ for non-convex objectives, the same as vanilla SGD [31, 37]. To the best of our knowledge, `AQ-SGD` is the first activation compression algorithm with rigorous theoretical analysis that shows a convergence rate of $O(1/\sqrt{T})$ (without relying on assumptions of unbiased gradient). **(Contribution 3)** We then show that `AQ-SGD` can be

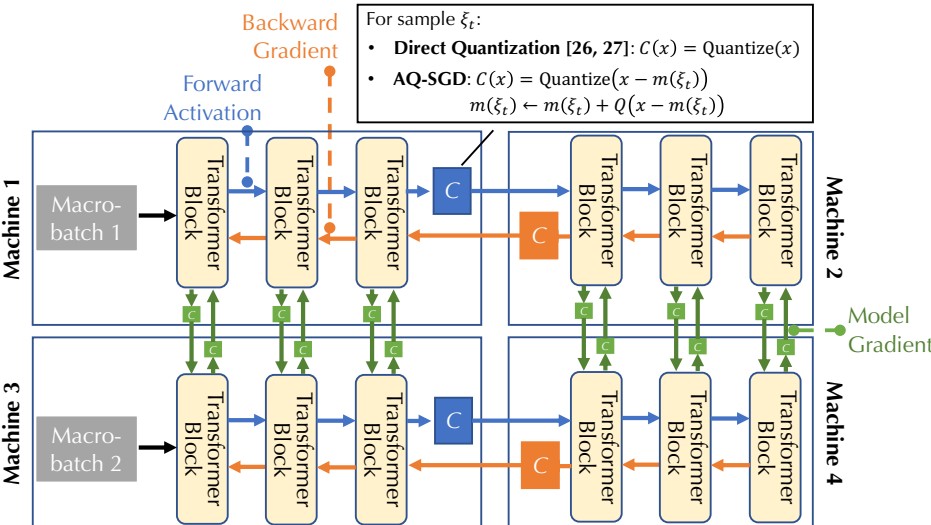

Figure 2: The communication pattern of training large language models with both data parallelism and pipeline model parallelism. $C$ denotes a compression module. The goal of this paper is to understand the design of $C$ for *forward activation* and *backward gradient*.

optimized and implemented efficiently[2], without adding additional end-to-end runtime overhead over non-compression and other compression schemes (it does require us to utilize more memory and SSD for storage of activations). **(Contribution 4)** We then conduct extensive experiments on sequence classification and language modeling datasets using DeBERTa-1.5B and GPT2-1.5B models, respectively. We show that `AQ-SGD` can aggressively quantize activations to 2-4 bits without sacrificing convergence performance, where direct quantization of activations fails to converge; in slow networks, `AQ-SGD` achieves up to $4.3\times$ end-to-end speedup. **(Contribution 5)** Last but not least, we also show that `AQ-SGD` can be combined with state-of-the-art gradient compression algorithms to enable "end-to-end communication compression": *All data exchanges between machines, including model gradients, forward activations, and backward gradients are quantized into lower precision.* This provides up to $4.9\times$ end-to-end speed-up, without sacrificing model quality.

## 2    Overview and Problem Formulation

Training large language models over multiple devices is a challenging task. Because of the vast number of parameters of the model and data examples, state-of-the-art systems need to combine different forms of parallelism. Figure 2 illustrates an example in which such a model is distributed over four machines: *(Pipeline Model Parallelism)* The model is partitioned onto Machine 1 and Machine 2 (similarly, Machine 3 and Machine 4), each of which is holding a subset of *layers*. To compute the gradient over the model using backpropagation, these two machines need to communicate the *activations* during the forward pass and the *gradient on activations* during the backward pass. *(Data Parallelism)* Machine 1 and Machine 3 (similarly, Machine 2 and Machine 4) process the same set of *layers* for different *macro-batches*. In this case, each of them will hold a replica of the same model. After the gradient over their model parameters are ready, they need to communicate the *model gradient*, usually by taking an average [15, 17, 18].

**Communication Compression for Forward Activations and Backward Gradients.**    In slow networks, the communication among all machines often becomes the bottleneck [37]. To improve the speed of training, one can conduct *lossy compression* of the data before they are communicated, illustrated as the $C$ module in Figure 2. When the model fits into a single machine, there have been intensive efforts on compressing the model gradient [5, 6, 7, 8]. However, when it comes to pipeline model parallelism, such compression techniques are less studied. In this paper, we focus on designing

---

[2]Our code is available at: `https://github.com/DS3Lab/AC-SGD`.

efficient communication compression algorithms to compress both forward activations and backward gradients. As we will show later, both can be compressed significantly with `AQ-SGD` without hurting the model quality. We also show that it is possible to combine `AQ-SGD` with state-of-the-art gradient compression techniques to enable the end-to-end compression scheme illustrated in Figure 2.

**Problem Formulation.** In this paper, we focus on the following technical problem. Note that, for the simplicity of notations, we present here the case where the model is partitioned onto $K = 2$ machines. `AQ-SGD` works for cases with $K > 2$: (1) in experiments, we consider $K = 8$, i.e., a single model is partitioned onto 8 machines; (2) in the supplementary material we provide the theoretical analysis for $K > 2$.

Given a distribution of samples $\mathcal{D}$, we consider the following optimization task:

$$\min_{x \in \mathbb{R}^d} \quad f(x) := \mathbb{E}_{\xi \sim \mathcal{D}} F(b(a(\xi, x^{(a)}), x^{(b)})), \tag{2.1}$$

where $F$ is a loss function, $a(-)$ and $b(-)$ correspond to two sets of *layers* of the model — $a(-)$ has model $x^{(a)}$ and $b(-)$ has model $x^{(b)}$. In Figure 2, Machine 1 would hold $x^{(a)}$ and Machine 2 would hold $x^{(b)}$. In the following, we call the machine that holds $x^{(a)}$ Machine $a$ and the machine that holds $x^{(b)}$ Machine $b$ . In the standard backpropagation algorithm, the communication between these two machines are as follows:

- Given a data sample $\xi$, Machine $a$ sends to Machine $b$ the forward activation: $a(\xi, x^{(a)})$
- Machine $b$ sends to Machine $a$ the backward gradient on $a(\xi, x^{(a)})$.

**Difficulties in Direct Quantization.** A natural way at compressing forward activations is to send, instead of $a(\xi, x^{(a)})$, a quantized version $m(\xi, x^{(a)}) = Q(a(\xi, x^{(a)}))$. This is the quantization scheme that state-of-the-art methods such as AC-GC [29] and TinyScript [30] use. Both AC-GC [29] and TinyScript [30] assume that gradient is unbiased when $m(\xi, x^{(a)})$ is an unbiased estimator of $a(\xi, x^{(a)})$. This enables their convergence rates of $\mathcal{O}(1/\sqrt{T})$. However, because of the non-linearity of $F$ and $b$ in a deep learning model with non-linear activation functions, an unbiased $m(\xi, x^{(a)})$ *will* lead to biases on the gradient. In Appendix, we will provide an example showing that such a gradient bias will hurt SGD convergence even for a very simple optimization problem. On the theory side, previous efforts on understanding gradient bias [38] have also shown that even bounded bias on gradient can impact the converges of SGD. As we will show later, empirically, this bias can indeed lead to suboptimal models under aggressive compression.

**Notation.** Throughout the paper we use the following definitions:

- $f^*$ is the optimal value of $f$.
- $N$ is the number of samples.
- $x_t = (x_t^{(a)}, x_t^{(b)})$ is the full model at iteration $t$.
- $\nabla f(\cdot)$ is the gradient of function $f$.
- $g_{\xi_t}(x_t) = \nabla F(\xi_t; x_t)$ is the stochastic gradient.
- $Q(\cdot)$ is the quantization function used to compress activations.
- $m(\cdot)$ is the message exchanged between $a$ and $b$ in the feed forward path.
- $\| \cdot \|$ denotes the $L_2$-norm.

## 3   `AQ-SGD`: Theoretical Analysis and System Implementations

In this section we present the `AQ-SGD`, with the goal to mitigate the above mentioned difficulties that appear in direct quantization of the activation functions.

### 3.1   `AQ-SGD` **Algorithm**

Algorithm 1 illustrates the `AQ-SGD` algorithm. The idea behind Algorithm 1 is simple — *instead of quantizing the activations directly, quantize the changes of activations for the same training example across epochs*. As illustrated in Algorithm 1, for iteration $t$ and the data sample $\xi_t$, *if* it is the first time

---

**Algorithm 1** `AQ-SGD` Algorithm

---

1: **Initialize:** $x_0$, learning rate $\gamma$, sub-network $a(-)$ weights $x^{(a)}$, sub-network $b(-)$ weights $x^{(b)}$,
   quantization function $Q$, array of previous messages $m$ initialized to 0
2: **for** t = 1, ..., T **do**
3:    Randomly sample $\xi_t$
4:    **if** $\xi_t$ **not** seen before **then**
5:       Set $m(\xi_t) = a(\xi_t, x_t^{(a)})$
6:    **else**
7:       Update $m(\xi_t) \leftarrow m(\xi_t) + Q\big(a(\xi_t, x_t^{(a)}) - m(\xi_t)\big)$
8:    **end if**
9:    // Machine $a$ sends $Q\big(a(\xi_t, x_t^{(a)}) - m(\xi_t)\big)$ to Machine $b$, which knows $m(\xi_t)$ through a local
      version of $m$
10:   Update $x_{t+1}^{(b)} \leftarrow x_t^{(b)} - \gamma \cdot \nabla_{x^{(b)}}(f \circ b)|_m$
11:   // Machine $b$ sends $Q(\nabla_a(f \circ b)|_m)$ to Machine $a$
12:   Update $x_{t+1}^{(a)} \leftarrow x_t^{(a)} - \gamma \cdot Q(\nabla_a(f \circ b)|_m) \cdot \nabla_{x^{(a)}} a$
13: **end for**
14: **Output:** $x = (x_T^{(a)}, x_T^{(b)})$

---

that $\xi_t$ is sampled, Machine $a$ communicates the full precision activations without any compression: $m(\xi_t) = a(\xi_t, x_t^{(a)})$ (Lines 4-5). Both machines will save $m(\xi_t)$ in a local buffer. If $\xi_t$ has been sampled in previous iterations, Machine $a$ communicates a quantized version:

$$Q(a(\xi_t, x_t^{(a)}) - m(\xi_t)),$$

where $m(\xi_t)$ was the previous message, stored in the local buffer. Both machines then update this local buffer:

$$m(\xi_t) \leftarrow m(\xi_t) + Q(a(\xi_t, x_t^{(a)}) - m(\xi_t)).$$

Machine $b$ then use $m(\xi_t)$ as the forward activations, compute backward gradients, and communicate a quantized version of the backward gradient to Machine $a$ (Line 11). We use

$$\delta_\xi = a(\xi_t, x_t^{(a)}) - m(\xi_t)$$

to denote the *message error* in sending the activations.

**Update Rules.** The above algorithm corresponds to the following update rules, at iteration $t$ with sample $\xi_t$:

$$x_{t+1}^{(a)} = x_t^{(a)} - \gamma \cdot Q(\nabla_a(f \circ b)|_{(m(\xi, x_t^{(a)}), x_t^{(b)})}) \cdot \nabla_{x^{(a)}} a|_{x_t^{(a)}},$$

$$x_{t+1}^{(b)} = x_t^{(b)} - \gamma \cdot \nabla_{x^{(b)}}(f \circ b)|_{(m(\xi, x_t^{(a)}), x_t^{(b)})},$$

where $\gamma$ is the learning rate, $\nabla_{x^{(b)}}(f \circ b)|_{(m(\xi, x_t^{(a)}), x_t^{(b)})}$ is the gradient on $x^{(b)}$ using the quantized forward activations $(m(\xi, x_t^{(a)})$, and $Q(\nabla_a(f \circ b)|_{(m(\xi, x_t^{(a)}), x_t^{(b)})})$ is the quantized backward gradient.

Setting $x_t = (x_t^{(a)}, x_t^{(b)})$, we can rephrase the update rule as

$$x_{t+1} = x_t - \gamma \cdot (g_\xi(x_t) + \Delta_\xi(x_t)),$$

where $g_\xi(x_t)$ is the stochastic gradient and $\Delta_\xi(x_t)$ is the *gradient error* introduced by communication compression. We have $\Delta_\xi = (\Delta_\xi^{(a)} + \Delta_\xi^{(Q)}, \Delta_\xi^{(b)})$ given by:

$$\Delta_\xi^{(Q)}(x_t) = Q(\nabla_a(f \circ b)|_{(m(\xi, x_t^{(a)}), x_t^{(b)})}) \cdot \nabla_{x^{(a)}} a|_{x_t^{(a)}} - \nabla_a(f \circ b)|_{(m(\xi, x_t^{(a)}), x_t^{(b)})} \cdot \nabla_{x^{(a)}} a|_{x_t^{(a)}},$$

$$\Delta_\xi^{(a)}(x_t) = \nabla_a(f \circ b)|_{(m(\xi, x_t^{(a)}), x_t^{(b)})} \cdot \nabla_{x^{(a)}} a|_{x_t^{(a)}} - \nabla_a(f \circ b \circ a)|_{(x_t^{(a)}, x_t^{(b)})}$$

$$\Delta_\xi^{(b)}(x_t) = \nabla_{x^{(b)}}(f \circ b)|_{(m(\xi, x_t^{(a)}), x_t^{(b)})} - \nabla_{x^{(b)}}(f \circ b)|_{(a(\xi, x_t^{(a)}), x_t^{(b)})},$$

where $\Delta_\xi^{(Q)}(x_t)$ is the error introduced by the gradient quantization in the backpropagation part, whilst $\Delta_\xi^{(a)}(x_t)$ and $\Delta_\xi^{(b)}(x_t)$ are the errors that the gradients of $a$ and $b$, respectively, inherit from the bias introduced in the forward pass.

## 3.2  Theoretical Analysis

We now prove the main theorem which states that, under some standard assumptions that are often used in the literature [31, 37], the convergence rate of `AQ-SGD` algorithm is $O(1/\sqrt{T})$ for non-convex objectives, same as vanilla SGD.

**Assumptions.**  We make several assumptions on the networks and the quantization. It is important to note is that we put no restrictions on either the message error $\delta_\xi$, nor the gradient error $\Delta_\xi$.

- **(A1: Lipschitz assumptions)** We assume that $\nabla f$, $\nabla(f \circ b)$ and $a$ are $L_f$, $L_{f \circ b}$-, and $\ell_a$-Lipschitz, respectively, recalling that a function $g$ is $L_g$-Lipschitz if

$$\|g(x) - g(y)\| \le L_g \|x - y\|, \qquad \forall x, \forall y.$$

  Furthermore, we assume that $a$ and $f \circ b$ have gradients bounded by $C_a$ and $C_{f \circ b}$, respectively, i.e. $\|\nabla a(x)\| \le C_a$, and $\|\nabla(f \circ b)(x)\| \le C_{f \circ b}$.

- **(A2: SGD assumptions)** We assume that the stochastic gradient $g_\xi$ is unbiased, i.e. $\mathbb{E}_\xi[g_\xi(x)] = \nabla f(x)$, for all $x$, and with bounded variance, i.e. $\mathbb{E}_\xi \|g_\xi(x) - \nabla f(x)\|^2 \le \sigma^2$, for all $x$.

**Theorem 3.1.** *Suppose that Assumptions A1, A2 hold, and consider an unbiased quantization function $Q(x)$ which satisfies that there exists $c_Q < \sqrt{1/2}$ such that $\mathbb{E}\|x - Q(x)\| \le c_Q \|x\|$, for all $x$.*[3] *Let $\gamma = \frac{1}{3(3L_f + C)\sqrt{T}}$ be the learning rate, where*

$$C = \frac{4 c_Q \ell_a (1 + C_a) L_{f \circ b} N}{\sqrt{1 - 2 c_Q^2}}.$$

*Then after performing $T$ updates one has*

$$\frac{1}{T} \sum_{t \in [T]} \mathbb{E} \|\nabla f(x_t)\|^2 \lesssim \frac{(C + L_f)(f(x_1) - f^*)}{\sqrt{T}} + \frac{\sigma^2 + (c_Q C_a C_{f \circ b})^2}{\sqrt{T}}. \tag{3.1}$$

We present the full proof of Theorem 3.1 in Appendix **??**, whereas here we explain the main intuition. The usual starting point in examining convergence rates is to use the fact that $f$ has $L_f$-Lipschitz gradient. It is well known that this implies

$$\gamma \langle \nabla f(x_t), g_{\xi_t}(x_t) \rangle + f(x_{t+1}) - f(x_t) \le -\langle \nabla f(x_t), \Delta_{\xi_t}(x_t) \rangle + \frac{\gamma^2 L_f}{2} \|g_{\xi_t}(x_t) + \Delta_{\xi_t}(x_t)\|^2.$$

After taking the expectation over all $\xi_t$ and summing over all $t = 1, \ldots, T$, we easily see that the key quantity to bound is $\sum_{t=1}^{T} \mathbb{E}\|\tilde{\Delta}_{\xi_t}(x_t)\|^2$, where $\tilde{\Delta}_{\xi_t}(x_t) = (\Delta_{\xi_t}^{(a)}(x_t), \Delta_{\xi_t}^{(b)}(x_t))$. On the other hand, the main object that we can control is the message error, $\delta_\xi$. Therefore, we first prove an auxiliary result which shows that $\|\tilde{\Delta}_{\xi_t}(x_t)\| \le (1 + C_a)\ell_a \|\delta_{\xi_t}(x_t)\|$, for all $t$. The key arguments for bounding $\delta_{\xi_t}$ closely follow the self-improving loop described in the introduction, and can be summarized as follows. Since we are compressing the information in such a way that we compare with all the accumulated differences, this allows us to unwrap the changes which appeared since the last time that we observed the current sample, in an iterative way. However, since these are gradient updates, they are bounded by the learning rate — as long as we have a quantization method that keeps enough information about the signal, we can recursively build enough saving throughout the process. In particular, the more stability we have in the process, the smaller the changes of the model and the compression error gets, further strengthening the stability.

**Tightness.**  The bound is tight with respect to quantization — setting $c_Q = 0$ (implying $C = 0$), i.e. quantization does not incur any loss, recovers the original original SGD convergence (cf. [31, 37]).

---

[3] Even for a very simple quantization function $Q(x) = \|x\| \cdot \lceil x/\|x\|\rfloor$, where $\lceil \cdot \rfloor$ denotes rounding to the closest $k/2^b$, stochastically, through a simple bound $c_Q = \sqrt{d}/2^b$, 6 bits suffice in a low-dimensional ($\sim 10^3$), 11 bits in a high-dimensional scenario ($\sim 10^6$), and 16 bits in a super-high-dimensional scenario ($\sim 10^9$). In practice, as we show in the experiments, we observe that for 2-4 bits are often enough for fine-tuning GPT-2 style model. This leaves interesting direction for future exploration as we expect a careful analysis of sparsity together with more advanced quantization functions can make this condition much weaker.

**Regularization and other optimizers.** Assuming A1 and A2 for $f$, and under further assumptions on $b$ and $\nabla_{x^{(b)}} b$, one can prove that the $L_2$-regularized loss $\tilde{f}(x) = f(x) + \frac{\lambda}{2}\|x\|^2$, which results in weight decay, satisfies Assumptions A1 and A2 with slightly weaker constants. We note that a theoretical analysis for other regularization methods or optimizers such as Adam [39], could be an independent study and represent an interesting line of future research.

### 3.3 System Implementations and Optimizations.

**Additional storage and communication.** `AQ-SGD` requires us to store, for each data example $\xi$, the quantized activation $m(\xi)$ in a local buffer in memory or SSD. For example, in GPT2-XL training, a simple calculation shows that we need an approximately extra 1TB storage. When using data parallelism, it reduces to 1TB / # parallel degree, but also incurs communication overhead if data is shuffled in every epoch. In addition, when the example $\xi$ is sampled again, $m(\xi)$ needs to be (1) loaded from this local buffer to the GPU, and (2) updated when a new value for $m(\xi)$ is ready.

**Optimization.** It is easy to implement and optimize the system such that this additional loading and updating step do not incur significant overhead on the end-to-end runtime. This is because of the fact that the GPU computation time for a forward pass is usually much longer than the data transfer time to load the activations — for GPT2-XL with 1.5 billion model parameters, a single forward pass on 6 layers require 44 ms, whereas loading $m(\xi)$ need 0.2 ms from memory and 12 ms from SSD. One can simply pre-fetch $m(\xi)$ right before the forward pass, and hide it within the forward pass of other data examples. Similarly, updating $m(\xi)$ can also be hidden in the backward computation. It is also simple to reduce the communication overhead by shuffling data only once or less frequently.

## 4 Evaluation

We demonstrate that `AQ-SGD` can significantly speed up fine-tuning large language models in slow networks. Specifically, we show: (1) on four standard benchmark tasks, `AQ-SGD` can tolerate aggressive quantization on the activations (2-4 bits) and backward gradients (4-8 bits), without hurting convergence and final loss, whereas direct quantization converges to a worse loss or even diverge, (2) in slow networks, `AQ-SGD` provide an end-to-end speed-up up to $4.3\times$, and (3) `AQ-SGD` can be combined with state-of-the-art gradient compression methods and achieve an end-to-end speed-up of up to $4.9\times$.

### 4.1 Experimental Setup

**Datasets and Benchmarks.** We consider both *sequence classification* and *language modeling* tasks with state-of-the-art foundation models. For sequence classification, we fine-tune a 1.5B parameter DeBERTa[4] on two datasets: QNLI and CoLA. For language modeling, we fine-tune the GPT2 model with 1.5B parameters[5] on two datasets: WikiText2 and arXiv abstracts. All datasets are publicly available and do not contain sensitive or offensive content. Detailed setup can be found in Appendix **??**.

**Distributed Cluster.** We conduct our experiments on AWS with 8-32 `p3.2xlarge` instances, each containing a V100 GPU. For a single pipeline, we partition a model onto 8 machines. When combined with data parallelism, we use 32 instances — data parallel degree is 4 and pipeline parallel degree is 8. By default, instances are interconnected with 10Gbps bandwidth. We simulate slow networks by controlling the communication bandwidth between instances using Linux traffic control.

**Baselines.** We compare with several strong baselines:

1. `FP32`: in which all communications are in 32 bit floating point without any compression.
2. `DirectQ` [29, 30]: in which activations and backward gradients are directly quantized.

We use a simple, uniform quantization scheme, which first normalizes a given vector into $[-1, 1]$ and quantize each number into a $b$-bit integers by uniforming partitioning the range $[-1, 1]$ into $2^b$ intervals [35]. Additional details of the configuration can be found in Appendix **??**.

---

[4]we use the v2-xxlarge checkpoint: `https://huggingface.co/microsoft/deberta-v2-xxlarge`.
[5]we use the extra large checkpoint: `https://huggingface.co/gpt2-xl`.

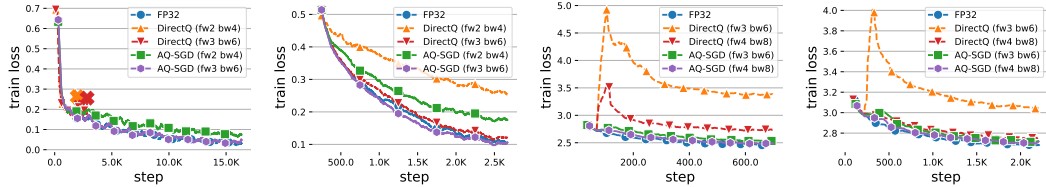

(a) QNLI, DeBERTa-1.5B    (b) CoLA, DeBERTa-1.5B    (c) WikiText2, GPT2-1.5B    (d) arXiv, GPT2-1.5B

Figure 3: Convergence (loss vs. # steps) of different approaches. $\times$ represents divergence.

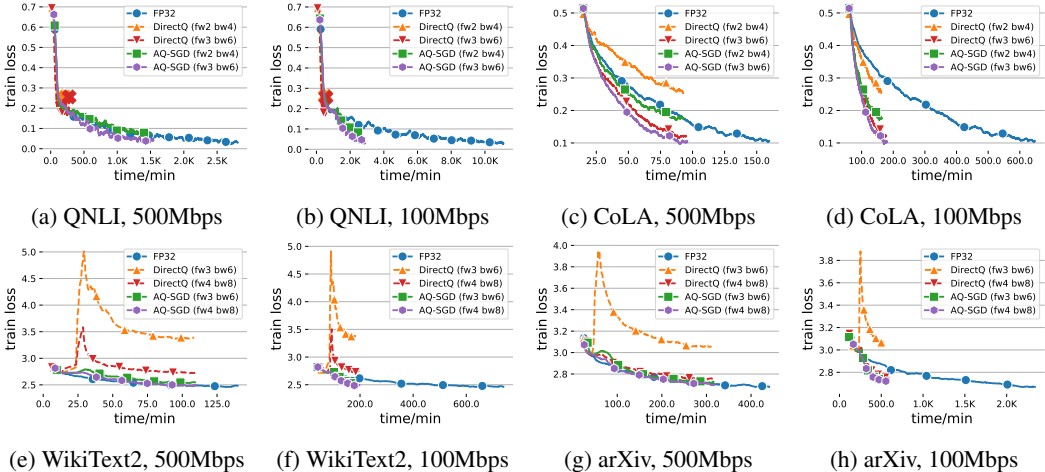

(a) QNLI, 500Mbps    (b) QNLI, 100Mbps    (c) CoLA, 500Mbps    (d) CoLA, 100Mbps

(e) WikiText2, 500Mbps    (f) WikiText2, 100Mbps    (g) arXiv, 500Mbps    (h) arXiv, 100Mbps

Figure 4: End-to-end training performance over different networks. $\times$ represents divergence.

**Hyperparameter Tuning.** We conduct careful tuning for all methods on all datasets. We perform grid search to choose learning rate from {2.5e-6, 3e-6, 5e-6, 1e-5} and macro-batch size from {32, 64, 96} for best model performance. We train all models using the Adam optimizer with weight decay.

## 4.2 Results

**Convergence.** We first compare the convergence behavior of different methods. For all compression methods, we try various settings: `fw`$x$ `bw`$y$ means that we use $x$ bits for forward activation and $y$ bits for backward gradients. Figure 3 shows the convergence behavior for the sequence classification and language modeling tasks. `FP32` converges fast since it does not introduce any compression errors. `DirectQ`, under aggressive quantization, can converge to a significantly worse model, or even diverge. This is not surprising, given the biases on model gradients that direct quantization introduced. On the other hand, `AQ-SGD` converges almost as fast as `FP32` in terms of number of training steps.

**End-to-End Runtime.** We show the end-to-end runtime of different methods under slow networks. As illustrated in Figure 4, `AQ-SGD` achieves a 4.3$\times$ end-to-end speed-up comparing with that of `FP32` (in terms of time to the same loss), illustrating the importance of communication compression in slow networks. Table 2 shows the training throughput and Table 3 shows the breakdown of our algorithm. We note that computation and communication can overlap, so the end-to-end time depends on the larger one of the two. Another interesting observation is that when the network is 100$\times$ slower (from 10Gbps to 100Mbps), the training is only about 1.18$\times$ slower! This is exciting — if `AQ-SGD` were to be deployed in a in real-world geo-distributed decentralized networks, the training throughput would be almost as fast as the performance inside a data center!

Moreover, `AQ-SGD` does not introduce significant runtime overhead over direct quantization. From Table 2, we see that `AQ-SGD` is essentially as efficient as direct quantization compression in throughput.

Table 2: Training Throughput of GPT2-1.5B. Others are similar and shown in Appendix.

| Network Bandwidth | FP32 | DirectQ | | AQ-SGD | |
|---|---|---|---|---|---|
| | | fw3 bw6 / fw4 bw8 | | fw3 bw6 / fw4 bw8 | |
| 10 Gbps | 3.8 | 4.0 | / 4.1 | 4.0 | / 4.0 |
| 1 Gbps | 3.2 | 4.0 | / 4.0 | 4.0 | / 3.9 |
| 500 Mbps | 2.7 | 3.9 | / 3.9 | 3.9 | / 3.9 |
| 300 Mbps | 1.8 | 3.9 | / 3.8 | 3.8 | / 3.8 |
| 100 Mbps | 0.5 | 3.5 | / 3.0 | 3.4 | / 3.0 |

Table 3: Breakdown of `AQ-SGD` (fw4 bw8) on GPT2-1.5B. We show the computation and communication time of each micro batch.

| Network Bandwidth | Forward pass | | Backward pass | |
|---|---|---|---|---|
| | comp. | comm. | comp. | comm. |
| 500 Mbps | 45 ms | 13 ms | 135 ms | 25 ms |
| 300 Mbps | 45 ms | 21 ms | 135 ms | 42 ms |
| 200 Mbps | 45 ms | 31 ms | 135 ms | 63 ms |
| 100 Mbps | 45 ms | 63 ms | 135 ms | 125 ms |

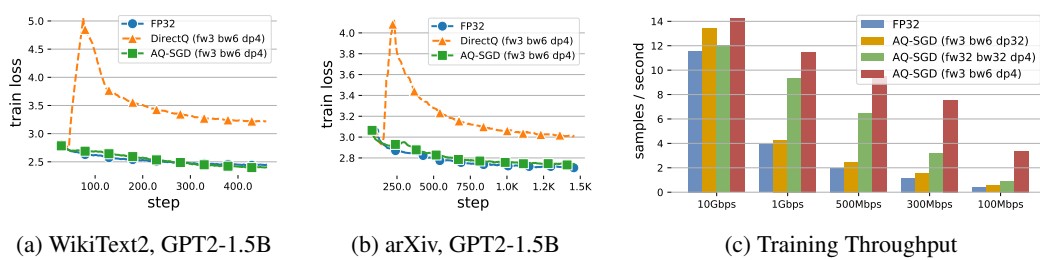

(a) WikiText2, GPT2-1.5B     (b) arXiv, GPT2-1.5B     (c) Training Throughput

Figure 5: Convergence and Throughput of `AQ-SGD` combined with gradient compression.

## 4.3   End-to-end Communication Compression: `AQ-SGD` + QuantizedAdam

`AQ-SGD` can also be combined with existing methods on gradient compression. This allows us to compress *all* communications during training. We combine `AQ-SGD` with QuantizedAdam [40], an error compensation-based gradient compression algorithm for data parallel training.

We quantize the forward activations with 3 bits, the backward gradient with 6 bits, and model gradient with 4 bits. Figure 5 illustrates the convergence and the training throughput under different network configuration. We see that `AQ-SGD` converges well when combined with QuantizedAdam (Figure 5(a, b)). On the other hand, `DirectQ`, when combined with gradient compression, converges to a much worse model. In terms of training throughput, with both activation and gradient compression, we can achieve up to 8.5× throughput improvement compared to the no-compression baseline (Figure 5(c)). We also see that *both* activation and gradient compression are important in terms of improving end-to-end training throughput — as illustrated in Figure 5(c), disabling any of them will lead to a much lower training throughput.

## 5   Related Work

**Distributed training of foundation models.** Modern distributed training of deep neural networks goes beyond data parallelism [24, 41, 42] due to the advance of the large-scale foundation models [20], such as BERT [21], GPT-3 [22], and CLIP[23]. Popular systems to support foundation model training include Megatron [24], Deepspeed [25], and Fairscale [26]. To scale out the training of large-scale models, pipeline parallelism (e.g., Gpipe[41], Pipedream[43, 44]) is a popular option, where the model is partitioned into multiple stages, different stages are allocated on different GPUs and the exchange of activations and gradients on activations goes through network communication.

**Communication compression of distributed learning.** Communication compression is an effective system relaxation for distributed training, especially in data parallelism [40, 45, 46, 47, 48, 49, 50, 51, 52, 53, 54, 55]. Popular techniques include quantization [5, 6, 7, 8], sparsification [56, 57, 58, 59], sketching [60, 61] and error compensation [27]. Recently, TinyScript [30] proposes to compress activations and gradients simultaneously.

**Sparse Learning for activation compression.** Sparse learning [30, 62, 63, 64, 65, 66, 67, 68, 69] has become increasingly popular for training neural networks, as it can significantly reduce the use of computation and memory while preserving the generalization of such models. In particular,

activation compression methods [32, 33, 34, 35, 36, 29] are proposed to reduce the memory footprint by adopting lossless [70, 71, 72] and lossy [73, 29, 74] compression techniques in the training of various deep neural networks (e.g., CNN[66, 75, 76], GNN[77]). These approaches usually compute the activation with *full precision* in forward propagation, adopt the compression method over the activation, and store the compressed version in DRAM for later use in backward propagation. Thus, compression does not introduce any error in forward propagation in contrast to the scenario of communicating compressed activation in the distributed setting.

**Delta-based compression.** Delta-based compression [78] is a classic solution to various system problems. Recent research has also used the property of spatial proximity within activation in neural network training based on empirical observations [79, 80, 74]. However, to our knowledge, no attempt has been made to consider the proximity of activation through training epochs to enable efficient compression with theoretical guarantee.

# 6 Conclusion

In this paper, we discuss how to adopt communication compression for activations in distributed learning. We proposed `AQ-SGD`, a novel activation compression algorithm for communication-efficient pipeline parallelism training over slow networks. `AQ-SGD` achieves $O(1/\sqrt{T})$ convergence rate for non-convex optimization without making assumptions on gradient unbiasedness. Our empirical study suggests that `AQ-SGD` can achieve up to $4.3\times$ speedup for pipeline parallelism. When combined with gradient compression in data parallelism, the end-to-end speed-up can be up to $4.9\times$.

# Acknowledgments

CZ and the DS3Lab gratefully acknowledge the support from the Swiss State Secretariat for Education, Research and Innovation (SERI) under contract number MB22.00036 (for European Research Council (ERC) Starting Grant TRIDENT 101042665), the Swiss National Science Foundation (Project Number 200021_184628, and 197485), Innosuisse/SNF BRIDGE Discovery (Project Number 40B2-0_187132), European Union Horizon 2020 Research and Innovation Programme (DAPHNE, 957407), Botnar Research Centre for Child Health, Swiss Data Science Center, Alibaba, Cisco, eBay, Google Focused Research Awards, Kuaishou Inc., Oracle Labs, Zurich Insurance, and the Department of Computer Science at ETH Zurich. CR gratefully acknowledges the support of NIH under No. U54EB020405 (Mobilize), NSF under Nos. CCF1763315 (Beyond Sparsity), CCF1563078 (Volume to Velocity), and 1937301 (RTML); ARL under No. W911NF-21-2-0251 (Interactive Human-AI Teaming); ONR under No. N000141712266 (Unifying Weak Supervision); ONR N00014-20-1-2480: Understanding and Applying Non-Euclidean Geometry in Machine Learning; N000142012275 (NEPTUNE); NXP, Xilinx, LETI-CEA, Intel, IBM, Microsoft, NEC, Toshiba, TSMC, ARM, Hitachi, BASF, Accenture, Ericsson, Qualcomm, Analog Devices, Google Cloud, Salesforce, Total, the HAI-GCP Cloud Credits for Research program, the Stanford Data Science Initiative (SDSI), and members of the Stanford DAWN project: Facebook, Google, and VMWare. The U.S. Government is authorized to reproduce and distribute reprints for Governmental purposes notwithstanding any copyright notation thereon. Any opinions, findings, and conclusions or recommendations expressed in this material are those of the authors and do not necessarily reflect the views, policies, or endorsements, either expressed or implied, of NIH, ONR, or the U.S. This work was done during Jue Wang's visit to ETH Zürich, which was supported by Key Research and Development Program of Zhejiang Province of China (No. 2021C01009) and Fundamental Research Funds for the Central Universities. The computation required in this work was provided by Together Computer (`https://together.xyz/`).

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
