# A  Proof of the Main Theorem

In this section we prove Theorem 3.1. The main idea comes from the "self-enforcing" dynamics described in the introduction of this work: *the more training stabilizes $\rightarrow$ the smaller the changes of the model across epochs $\rightarrow$ the smaller the changes of activations for the same training example across epochs $\rightarrow$ the smaller the compression error using the same #bits $\rightarrow$ training stabilizes more.*

We start by providing two auxiliary results. The first one connects the message error and the gradient error.

**Lemma A.1.** *For every sample $\xi$, one has*

$$\|\Delta_\xi^{(Q)}(x)\| \le c_Q C_a C_{f \circ b},$$

*and*

$$\|\tilde{\Delta}_\xi(x)\| \le (1 + C_a) L_{f \circ b} \|\delta_\xi(x)\|,$$

*where $\tilde{\Delta}_\xi(x) = (\Delta_\xi^{(a)}(x), \Delta_\xi^{(b)})$.*

PROOF:   Note that

$$\|\Delta_\xi^{(Q)}(x_t)\| = \|\nabla_x a(\xi, x)_{x = x_t^{(a)}}\| \cdot \|Q(\nabla_x (f \circ b)(x, x_t^{(b)})_{x = m(\xi, x_t^{(a)})}) - \nabla_x (f \circ b)(x, x_t^{(b)})_{x = m(\xi, x_t^{(a)})}\|$$

$$\le C_a c_Q \|\nabla_x (f \circ b)(x, x_t^{(b)})_{x = m(\xi, x_t^{(a)})}\| \le c_Q C_a C_{f \circ b},$$

$$\|\Delta_\xi^{(a)}(x_t)\| = \|\nabla_x a(\xi, x)_{x = x_t^{(a)}}\| \cdot \|\nabla_x (f \circ b)(x, x_t^{(b)})_{x = m(\xi, x_t^{(a)})} - \nabla_x (f \circ b)(x, x_t^{(b)})_{x = a(\xi, x_t^{(a)})}\|$$

$$\le C_a L_{f \circ b} \|(m(\xi, x_t^{(a)}), x_t^{(b)}) - (a(\xi, x_t^{(a)}), x_t^{(b)})\| = C_a L_{f \circ b} \|\delta_\xi(x_t)\|,$$

$$\|\Delta_\xi^{(b)}(x_t)\| = \|\nabla_y (f \circ b)(m(\xi, x_t^{(a)}), y)_{y = x_t^{(b)}} - \nabla_y (f \circ b)(a(\xi, x_t^{(a)}), y)_{y = x_t^{(b)}}\|$$

$$\le L_{f \circ b} \|(m(\xi, x_t^{(a)}), x_t^{(b)}) - (a(\xi, x_t^{(a)}), x_t^{(b)})\| = L_{f \circ b} \|\delta_\xi(x_t)\|,$$

which together with $\|\tilde{\Delta}_\xi(x_t)\| = \|\Delta_\xi^{(a)}(x_t)\| + \|\Delta_\xi^{(b)}(x_t)\|$ yields the claim. ∎

We now prove that the message error can be efficiently bounded by the true gradient.

**Lemma A.2.** *For $C' = \frac{18 c_Q^2 l_a^2 N^2}{1 - 2c_Q^2}$, one has*

$$\frac{1}{T} \sum_{t \in [T]} \mathbb{E} \|\delta_{\xi_t}(x_t)\|^2 \le C' \gamma^2 \cdot \left( \frac{1}{T} \sum_{t \in [T]} \mathbb{E} \|\nabla f(x_t)\|^2 + \sigma^2 + (c_Q C_a C_{f \circ b})^2 \right).$$

PROOF:   Let $\xi$ be a fixed sample. To simplify the exposition, we abuse the notation slightly by $a(x) = a(\xi, x)$, $m(x) = m(\xi, x)$. Let $T(\xi)$ be the number of realizations of $\xi$ before time $T$. Using the definition of $\delta_\xi$ (recalling that $\delta_\xi(x_{t_1(\xi)}) = 0$, since in the first iteration we send the correct signal), we have

$$\sum_{k=1}^{T(\xi)} \|\delta_\xi(x_{t_k(\xi)})\|^2 = \sum_{k=2}^{T(\xi)} \|a(x_{t_k(\xi)}) - m(x_{t_k(\xi)})\|^2$$

$$= \sum_{k=2}^{T(\xi)} \|a(x_{t_k(\xi)}) - m(x_{t_{k-1}(\xi)}) - Q(a(x_{t_k(\xi)}) - m(x_{t_{k-1}(\xi)}))\|^2$$

$$\{\|x - Q(x)\| \le c_Q \|x\|\} \qquad \le c_Q^2 \sum_{k=2}^{T(\xi)} \|a(x_{t_k(\xi)}) - a(x_{t_{k-1}(\xi)}) + \delta_\xi(x_{t_{k-1}(\xi)})\|^2$$

$$\{(\alpha + \beta)^2 \le 2\alpha^2 + 2\beta^2\} \qquad \le 2c_Q^2 \sum_{k=2}^{T(\xi)} \|a(x_{t_k(\xi)}) - a(x_{t_{k-1}(\xi)})\|^2 + 2c_Q^2 \sum_{k=2}^{T(\xi)} \|\delta_\xi(x_{t_{k-1}(\xi)})\|^2$$

$$\{a \text{ is } \ell_a\text{-Lipschitz}\} \qquad \le 2c_Q^2 \ell_a^2 \sum_{k=2}^{T(\xi)} \|x_{t_k(\xi)} - x_{t_{k-1}(\xi)}\|^2 + 2c_Q^2 \sum_{k=2}^{T(\xi)} \|\delta_\xi(x_{t_{k-1}(\xi)})\|^2.$$

Transferring the $\delta_\xi$ part to the LHS, and noting that between every two realizations of $\xi$ at times $t_{k-1}(\xi)$ and $t_k(\xi)$, we can follow updates for $t = t_{k-1}(\xi), \ldots, t_k(\xi) - 1$, we get

$$(1 - 2c_Q^2) \sum_{k=1}^{T(\xi)} \|\delta_\xi(x_{t_k(\xi)})\|^2 \leq \gamma^2 \cdot (2c_Q^2 \ell_a^2) \sum_{k=1}^{T(\xi)} \left\| \sum_{t=t_{k-1}(\xi)}^{t_k(\xi)-1} (g_{\xi_t}(x_t) + \Delta_{\xi_t}(x_t)) \right\|^2$$

$$\{\text{Cauchy-Schwarz}\} \quad \leq \gamma^2 \cdot (2c_Q^2 \ell_a^2) \sum_{k=1}^{T(\xi)} (t_k(\xi) - t_{k-1}(\xi)) \sum_{t=t_{k-1}(\xi)}^{t_k(\xi)-1} \|g_{\xi_t}(x_t) + \Delta_{\xi_t}(x_t)\|^2$$

$$= \gamma^2 \cdot (2c_Q^2 \ell_a^2) \sum_{t \in [T]} \omega_\xi(t) \cdot \|g_{\xi_t}(x_t) + \Delta_{\xi_t}(x_t)\|^2,$$

where $\omega \colon [T] \to \{0, 1, \ldots\}$ is defined by $\omega_\xi(t) = t_k(\xi) - t_{k-1}(\xi)$, if $t \in [t_{k-1}(\xi), t_k(\xi) - 1]$, and $\omega_\xi(t) = 0$, if $t > t_{T(\xi)}(\xi)$. We note that for two different samples $\xi$ and $\xi'$, the sums on the LHS are disjoint. Therefore, summing over all samples $\xi$ and taking the expectation over $\xi$ and all $\xi_t$ yields

$$(1 - 2c_Q^2) \cdot \frac{1}{T} \sum_{t \in [T]} \mathbb{E}\|\delta_{\xi_t}(x_t)\|^2 \leq \gamma^2 \cdot (2c_Q^2 \ell_a^2) \cdot \frac{1}{T} \sum_{t \in [T]} \mathbb{E}\|g_{\xi_t}(x_t) + \Delta_{\xi_t}(x_t)\|^2 \cdot N \cdot \mathbb{E}[\omega_\xi(t)]$$

$$\leq \gamma^2 \cdot (2c_Q^2 \ell_a^2 N^2) \cdot \frac{1}{T} \sum_{t \in [T]} \mathbb{E}\|g_{\xi_t}(x_t) + \Delta_{\xi_t}(x_t)\|^2,$$

since $\mathbb{E}[\omega_\xi(t)] \leq N$. Applying $\|g_{\xi_t}(x_t) + \Delta_{\xi_t}(x_t)\|^2 \leq 3\|g_{\xi_t}(x_t)\|^2 + 3\|\Delta_{\xi_t}^{(Q)}(x_t)\|^2 + 3\|\tilde{\Delta}_{\xi_t}(x_t)\|^2$, bounded variance $\mathbb{E}\|g_\xi(x) - \nabla f(x)\|^2 \leq \sigma^2$, and Lemma A.1, we get

$$\left(1 - 2c_Q^2 - \gamma^2 \cdot 6c_Q^2 \ell_a^2 N^2 (1 + C_a)^2 L_{f \circ b}^2\right) \cdot \frac{1}{T} \sum_{t \in [T]} \mathbb{E}\|\delta_{\xi_t}(x_t)\|^2$$

$$\leq \gamma^2 \cdot (6c_Q^2 \ell_a^2 N^2) \left(2\sigma^2 + 2 \cdot \frac{1}{T} \sum_{t \in [T]} \mathbb{E}\|\nabla f(x_t)\|^2 + (c_Q C_a C_{f \circ b})^2\right),$$

which implies

$$\left(1 - 2c_Q^2 - \gamma^2 \cdot \frac{3}{8} \cdot C^2 \cdot (1 - 2c_Q^2)\right) \cdot \frac{1}{T} \sum_{t \in [T]} \mathbb{E}\|\delta_{\xi_t}(x_t)\|^2$$

$$\leq \gamma^2 \cdot (12c_Q^2 \ell_a^2 N^2) \left(\sigma^2 + \frac{1}{T} \sum_{t \in [T]} \mathbb{E}\|\nabla f(x_t)\|^2 + (c_Q C_a C_{f \circ b})^2\right),$$

Recalling the definitions of $C$ and $\gamma$, and the fact that $\gamma \cdot C < \frac{1}{3}$, we can simplify the LHS to get

$$(1 - 2c_Q^2) \cdot \frac{1}{T} \sum_{t \in [T]} \mathbb{E}\|\delta_{\xi_t}(x_t)\|^2 \leq \gamma^2 \cdot (12 \cdot \frac{24}{23} \cdot c_Q^2 \ell_a^2 N^2) \left(\sigma^2 + \frac{1}{T} \sum_{t \in [T]} \mathbb{E}\|\nabla f_{\xi_t}(x_t)\|^2 + (c_Q C_a C_{f \circ b})^2\right),$$

yielding the claim. $\blacksquare$

We are ready to prove the main result, with a learning rate of

$$\gamma = \frac{1}{3(3L_f + C)\sqrt{T}}.$$

PROOF OF THEOREM 3.1: Since $f$ has $L_f$-Lipschitz gradient, we know that

$$f(x_{t+1}) - f(x_t) \leq -\gamma \cdot \langle \nabla f(x_t), g_{\xi_t}(x_t) + \Delta_{\xi_t}(x_t) \rangle + \frac{\gamma^2 L_f}{2} \|g_{\xi_t}(x_t) + \Delta_{\xi_t}(x_t)\|^2.$$

Since the quantization is unbiased, implying $\mathbb{E}_Q[\Delta_\xi^{(Q)}(x)] = 0$, taking the expectation with respect to the quantization yields

$$\mathbb{E}_Q[f(x_{t+1})] - \mathbb{E}_Q[f(x_t)]$$
$$\leq -\gamma \mathbb{E}_Q \langle \nabla f(x_t), g_{\xi_t}(x_t) + \tilde{\Delta}_{\xi_t}(x_t) \rangle + \frac{3\gamma^2 L_f}{2} \left( \|g_{\xi_t}(x_t)\|^2 + \|\tilde{\Delta}_{\xi_t}(x_t)\|^2 + \mathbb{E}_Q \|\Delta_{\xi_t}^{(Q)}\|^2 \right),$$

where $\tilde{\Delta}_{\xi_t}(x_t) = (\Delta_{\xi_t}^{(a)}, \Delta_{\xi_t}^{(b)})$, and we used $(\alpha + \beta + \rho)^2 \leq 3\alpha^2 + 3\beta^2 + 3\rho^2$.

Taking the expectation over $\xi_t$ (simplifying the notation of $\mathbb{E}_Q \mathbb{E}_\xi$ to simply $\mathbb{E}$), and recalling that $\mathbb{E}[g_{\xi_t}(x_t)] = \nabla f(x_t)$, we can bound the first two terms of the RHS by

$$- \gamma \mathbb{E} \langle \nabla f(x_t), g_{\xi_t}(x_t) + \tilde{\Delta}_{\xi_t}(x_t) \rangle + \frac{3}{4} \gamma^2 L_f \mathbb{E} \|g_{\xi_t}(x_t) + \tilde{\Delta}_{\xi_t}(x_t)\|^2$$

$$\leq -\frac{\gamma}{2} \mathbb{E} \|\nabla f(x_t)\|^2 + \frac{\gamma}{2} \mathbb{E} \|\tilde{\Delta}_{\xi_t}(x_t)\|^2 + \frac{3}{2} \gamma^2 L_f \mathbb{E} \left( \|g_{\xi_t}(x_t)\|^2 + \|\tilde{\Delta}_{\xi_t}(x_t)\|^2 \right) \quad \{\alpha \cdot \beta \leq \frac{1}{2}(\alpha^2 + \beta^2)\}$$

$$\leq \left( -\frac{\gamma}{2} + 3\gamma^2 L_f \right) \mathbb{E} \|\nabla f(x_t)\|^2 + \left( \frac{\gamma}{2} + \frac{3}{2} \gamma^2 L_f \right) \mathbb{E} \|\tilde{\Delta}_{\xi_t}(x_t)\|^2 + 3\gamma^2 L_f \sigma^2 \quad \{\mathbb{E}\|g_{\xi_t} - \nabla f\|^2 \leq \sigma^2\}$$

$$\leq \left( -\frac{\gamma}{2} + 3\gamma^2 L_f \right) \mathbb{E} \|\nabla f(x_t)\|^2$$
$$+ \left( \frac{\gamma}{2} + \frac{3}{2} \gamma^2 L_f \right) (1 + C_a)^2 L_{f \circ b}^2 \mathbb{E} \|\delta_{\xi_t}(x_t)\|^2 + 3\gamma^2 L_f \sigma^2. \quad \{\text{Lemma A.2}\}$$

Reorganizing the terms, summing over all $t \in [T]$ and dividing by $T$ yields

$$\gamma \cdot \left( \frac{1}{2} - 3\gamma L_f \right) \cdot \frac{1}{T} \sum_{t \in [T]} \mathbb{E} \|\nabla f(x_t)\|^2 \leq \frac{f(x_1) - \mathbb{E}[f(x_{T+1})]}{T} + \gamma \cdot C'' \cdot \frac{1}{T} \sum_{t \in [T]} \mathbb{E} \|\delta_{\xi_t}(x_t)\|^2$$
$$+ \gamma^2 L_f (3\sigma^2 + \frac{3}{2} \cdot (c_Q C_a C_{f \circ b})^2),$$

where

$$C'' = \left( \frac{1}{2} + \frac{3}{2} \gamma L_f \right) (1 + C_a)^2 L_{f \circ b}^2 < (1 + C_a)^2 L_{f \circ b}^2, \tag{A.1}$$

by the definition of $\gamma$. Applying Lemma A.2 and regrouping the terms now yields

$$\gamma \cdot \left( \frac{1}{2} - 3\gamma L_f - \gamma^2 \cdot C' C'' \right) \cdot \frac{1}{T} \sum_{t \in [T]} \|\nabla f(x_t)\|^2$$
$$\leq \frac{f(x_1) - \mathbb{E}[f(x_{T+1})]}{T} + \gamma^2 (\gamma \cdot C' C'' + 3L_f) \cdot (\sigma^2 + (c_Q C_a C_{f \circ b})^2).$$

Noting that $C' C'' < C^2$ and recalling that $\gamma C < 1/3$ and $\gamma L_f < 1/9$, we get

$$\gamma \cdot \left( \frac{1}{2} - \gamma(3L_f + C) \right) \cdot \frac{1}{T} \sum_{t \in [T]} \|\nabla f_{\xi_t}(x_t)\|^2$$
$$\leq \frac{f(x_1) - \mathbb{E}[f(x_{T+1})]}{T} + \gamma^2 (3L_f + C) \cdot (\sigma^2 + (c_Q C_a C_{f \circ b})^2).$$

Since $\gamma \cdot (3L_f + C) = \frac{1}{3\sqrt{T}} \leq \frac{1}{3}$, the LHS coefficent is at least $\gamma/6$, so dividing by $\gamma/6$ now yields the claim by substituting $\gamma$ with $\frac{1}{3(3L_f + C)\sqrt{T}}$. ∎

## A.1 Theoretical analysis when $K > 2$

In this section we sketch how one can generalize the already provided theoretical analysis for the $K = 2$ case.

Instead of Machines $a$ and $b$, we now suppose that we have a stack $a_1, \ldots, a_K$ of $K$ Machines such that every pair $a_i, a_{i+1}$ communicates a compressed message, denoted by $m_i$. We simplify the notation of the complete model by $x = (x^{(1)}, \ldots, x^{(K)})$, and, for a sample $\xi$, further denote

$$\overline{a}^{(i)}(\xi, x_t) = a_i(a_{i-1}(\ldots (a_1(\xi, x_t^{(1)}), x_t^{(2)}), \ldots x_t^{(i)})),$$
$$\overline{m}^{(i)}(\xi, x_t) = m_i(m_{i-1}(\ldots (m_1(\xi, x_t^{(1)}), x_t^{(2)}), \ldots x_t^{(i)})),$$

for $i \in [K]$. As in Algorithm 1, for iteration $t$ and the data sample $\xi_t$, *if* it is the first time that $\xi_t$ is sampled, Machine $a_i$ communicates to Machine $a_{i+1}$ the full precision activations without any compression: $m_i(\xi_t) = \overline{a}^{(i)}(\xi_t, x_t)$. If $\xi_t$ has been sampled in previous iterations, Machine $a_i$ communicates a compressed version:

$$Q(a_i(\overline{m}^{(i-1)}(\xi_t), x_t^{(i)}) - m_i(\xi_t)),$$

where $m_i(\xi_t)$ was the previous message, stored in the local buffer. Both machines then update this local buffer:
$$m_i(\xi_t) \leftarrow m_i(\xi_t) + Q(a_i(\overline{m}^{(i-1)}(\xi_t), x_t^{(i)}) - m_i(\xi_t)).$$

Machine $a_{i+1}$ then uses $m_i(\xi_t)$ as the forward activations. Upon receiving backward gradients from Machine $a_{i+2}$, it computes backward gradients, and communicates a quantized version of the backward gradient to Machine $a_i$. We use

$$\delta_\xi^{(i)} = \overline{a}^{(i)}(\xi, x_t) - \overline{m}^{(i)}(\xi, x_t)$$

to denote the *message error* of $i$-th machine in sending the activations (accumulated also through messages in previous pairs), and denote the *total message error* by $\delta_\xi = (\delta_\xi^{(1)}, \ldots, \delta_\xi^{(K-1)})$.

**Update Rules.** We can now generalize the update rule for $a$ and $b$ to

$$x_{t+1}^{(K)} = x_t^{(K)} - \gamma \cdot \nabla_{x^{(K)}}(f \circ a_K)\big|_{(\overline{m}^{(K-1)}(\xi_t), x_t^{(K)})},$$
$$x_{t+1}^{(i)} = x_t^{(i)} - \gamma \cdot Q(\nabla_{a_i}(f \circ a_K \circ \ldots \circ a_{i+1})\big|_{(\overline{m}^{(i)}(\xi_t), x_t^{(i+1)}, \ldots, x_t^{(K)})}) \cdot \nabla_{x^{(i)}} a_i\big|_{(\overline{m}^{(i-1)}(\xi_t), x_t^{(i)})},$$

for $i = 1, \ldots, K-1$, where $\gamma$ is the learning rate. We can rephrase the update rule as

$$x_{t+1} = x_t - \gamma \cdot (g_\xi(x_t) + \Delta_\xi(x_t)),$$

where $g_\xi(x_t)$ is the stochastic gradient and $\Delta_\xi(x_t)$ is the *total gradient error* introduced by communication compression. We have $\Delta_\xi = (\Delta_\xi^{(1)} + \Delta_\xi^{(Q,1)}, \ldots, \Delta_\xi^{(K-1)} + \Delta_\xi^{(Q,K-1)}, \Delta_\xi^{(K)})$, given by:

$$\Delta_\xi^{(Q,i)}(x_t) = Q(\nabla_{a_i}(f \circ a_K \circ \ldots \circ a_{i+1})\big|_{(\overline{m}^{(i)}(\xi_t), x_t^{(i+1)}, \ldots, x_t^{(K)})}) \cdot \nabla_{x^{(i)}} a_i\big|_{(\overline{m}^{(i-1)}(\xi_t), x_t^{(i)})}$$
$$- \nabla_{a_i}(f \circ a_K \circ \ldots \circ a_{i+1})\big|_{(\overline{m}^{(i)}(\xi_t), x_t^{(i+1)}, \ldots, x_t^{(K)})} \cdot \nabla_{x^{(i)}} a_i\big|_{(\overline{m}^{(i-1)}(\xi_t), x_t^{(i)})},$$

$$\Delta_\xi^{(i)}(x_t) = \nabla_{a_i}(f \circ a_K \circ \ldots \circ a_{i+1})\big|_{(\overline{m}^{(i)}(\xi_t), x_t^{(i+1)}, \ldots, x_t^{(K)})} \cdot \nabla_{x^{(i)}} a_i\big|_{(\overline{m}^{(i-1)}(\xi_t), x_t^{(i)})}$$
$$- \nabla_{a_i}(f \circ a_K \circ \ldots \circ a_{i+1})\big|_{(\overline{a}^{(i)}(\xi_t), x_t^{(i+1)}, \ldots, x_t^{(K)})} \cdot \nabla_{x^{(i)}} a_i\big|_{(\overline{a}^{(i-1)}(\xi_t), x_t^{(i)})},$$

$$\Delta_\xi^{(K)}(x_t) = \nabla_{x^{(K)}}(f \circ a_K)\big|_{(\overline{m}^{(K-1)}(\xi_t), x_t^{(K)})} - \nabla_{x^{(K)}}(f \circ a_K)\big|_{(\overline{a}^{(K-1)}(\xi_t), x_t^{(K)})},$$

for all $i = 1, \ldots, K-1$.

**Generalized Assumptions.** In order to state the analogue of Theorem 3.1 for $K > 2$, we need to define the corresponding assumptions with respect to Lipschitz properties (whereas Assumption GA2 is here only for completeness, being the same as A2).

- **(GA1: Lipschitz assumptions)** We assume that
  - $f$ had $L_f$-Lipschitz gradient,
  - $f \circ a_K \circ \ldots \circ a_{i+1}$ has $L_{f \circ a_K \circ \ldots \circ a_{i+1}}$-Lipschitz gradient, and has gradient bounded by $C_{f \circ a_K \circ \ldots \circ a_{i+1}}$ for all $i = 1, \ldots, K-1$,
  - $a_i$ is $\ell_{a_i}$-Lipschitz, and has gradient bounded by $C_{a_i}$, for all $i = 1, \ldots, K$.

- **(GA2: SGD assumptions)** We assume that the stochastic gradient $g_\xi$ is unbiased, i.e. $\mathbb{E}_\xi[g_\xi(x)] = \nabla f(x)$, for all $x$, and with bounded variance, i.e. $\mathbb{E}_\xi\|g_\xi(x) - \nabla f(x)\|^2 \le \sigma^2$, for all $x$.

We have the following analogue of Theorem 3.1.

**Theorem A.3.** *Suppose that Assumptions GA1, GA2 hold, and let*

$$\tilde{C} = \sqrt{\sum_{i=1}^{K-1} C_{a_i}^2 C_{f \circ a_K \circ \ldots \circ a_{i+1}}^2}.$$

*Consider an unbiased quantization function $Q(x)$ which satisfies that there exists $c_Q < \sqrt{1/2}$ such that $\mathbb{E}\|x - Q(x)\| \le c_Q\|x\|$, for all $x$. Then there exists a constant $C$ that depends only on the constants defined above and on $\sqrt{K}$ and $N$, such that for the learning rate $\gamma$ proportional to $\frac{1}{(C+L_f)\sqrt{T}}$, after performing $T$ updates one has*

$$\frac{1}{T}\sum_{t\in[T]} \mathbb{E}\|\nabla f(x_t)\|^2 \lesssim \frac{(C+L_f)(f(x_1) - f^*)}{\sqrt{T}} + \frac{\sigma^2 + (c_Q\tilde{C})^2}{\sqrt{T}}. \tag{A.2}$$

Instead of performing a tedious job of rewriting the proof of the $K = 2$ case with inherently more constant-chasing, we will simply sketch the differences with respect to the proof of Theorem 3.1. First we note that, having analogous assumptions as in the case when $K = 2$, we can easily prove the following analogue of Lemma A.1.

**Lemma A.4.** *Let*

$$\Delta_\xi^{(Q)} = (\Delta_\xi^{(Q,1)}, \ldots, \Delta_\xi^{(Q,K-1)}).$$

*For every sample $\xi$, one has*

$$\|\Delta_\xi^{(Q,i)}(x)\| \le c_Q C_{a_i} C_{f \circ a_K \circ \ldots \circ a_{i+1}}, \qquad i = 1, \ldots, K-1,$$

$$\|\Delta_\xi^{(i)}(x)\| \le C_{a_i} L_{f \circ a_K \circ \ldots \circ a_{i+1}} \|\delta_\xi^{(i)}\| + C_{f \circ a_K \circ \ldots \circ a_{i+1}} L_{a_i} \|\delta_\xi^{(i-1)}\|, \qquad i = 1, \ldots, K-1,$$

*and*

$$\|\Delta_\xi^{(K)}(x)\| \le L_{f \circ a_K} \|\delta_\xi^{(K-1)}\|,$$

*implying $\|\Delta_\xi^{(Q)}\| \le \tilde{C} c_Q$.*

Comparing this with Lemma A.1, we see that for $\tilde{\Delta}_\xi = (\Delta_\xi^{(1)}, \ldots, \Delta_\xi^{(K)})$, we now have an additional term that depends on $\delta_\xi^{(i-1)}$. However, since in the proof of Theorem 3.1 we only rely on $\|\Delta_\xi\|^2$, we can proceed even with a crude bound

$$\|\tilde{\Delta}_\xi\|^2 = \sum_{i=1}^K \|\Delta_\xi^{(i)}\|^2$$

$$\le (1 + 2C_{a_{K-1}}^2) L_{f \circ a_K}^2 \|\delta_\xi^{(K-1)}\|^2 + 2\sum_{i=1}^{K-2}(C_{a_{K-2}}^2 L_{f \circ a_K \circ \ldots \circ a_{i+1}}^2 + C_{f \circ a_K \circ \ldots \circ a_{i+2}}^2 L_{a_{i+1}}^2)\|\delta_\xi^{(i)}\|^2$$

$$\le \underbrace{2K \cdot \max\left\{(1 + 2C_{a_{K-1}}^2) L_{f \circ a_K}^2, \max_{i\in[K-2]}\{C_{a_{K-2}}^2 L_{f \circ a_K \circ \ldots \circ a_{i+1}}^2 + C_{f \circ a_K \circ \ldots \circ a_{i+2}}^2 L_{a_{i+1}}^2\}\right\}}_{C_1} \cdot \|\delta_\xi\|^2.$$

Mimicking closely the steps of Lemma A.2, separately for each $i$ and the summing over all $i$ via $\|\ell_a\|^2 = \sum_{i=1}^K \|\ell_{a_i}\|^2$, one can straightforwardly yield the following analogue of Lemma A.2.

**Lemma A.5.** *For $C' = K \cdot \frac{36 c_Q^2 \|l_a\|^2 N^2 C_1}{1 - 2c_Q^2}$, where $l_a = (l_{a_1}, \ldots, l_{a_K})$, one has*

$$\frac{1}{T}\sum_{t\in[T]} \mathbb{E}\|\delta_{\xi_t}(x_t)\|^2 \le C'\gamma^2 \cdot \left(\frac{1}{T}\sum_{t\in[T]} \mathbb{E}\|\nabla f(x_t)\|^2 + \sigma^2 + (c_Q\tilde{C})^2\right).$$

Theorem A.3 can now be proved by repeating the steps of the proof of Theorem 3.1, carefully substituting Lemma A.1 by Lemma A.4, and Lemma A.2 by Lemma A.5.

| Dataset | # labels | # train samples | Task description |
|---|---|---|---|
| QNLI | 2 | 105K question-paragraph pairs | natural language inference |
| CoLA | 2 | 8.6K sentences | linguistic acceptability |
| WikiText2 | - | 2M tokens | language modeling |
| arXiv | - | 7M tokens | language modeling |

Table 4: Dataset statistics

## B  Benchmark Dataset

The data statistics can be found in Table 4. We fine-tune DeBERTa on QNLI and CoLA datasets. The QNLI task is to determine whether the context sentence contains the answer to the question. The CoLA task aims to detect whether a given sequence of tokens is a grammatical English sentence. In addition, we fine-tune GPT2 on the WikiText2 training set. We also collect 30K arXiv abstracts in 2021 to fine-tune GPT2. Neither corpus is included in the GPT2 pre-training data set.

## C  Training Task Setup and Hyper-parameter Tuning

**Sequence Classification.** We use the AdamW optimizer to fine-tune the model for 10 epochs. Specifically, for QNLI, we set the learning rate to 3.0e-6, learning rate warm-up steps to 1000, max sequence length to 256, the macro-batch size to 64 and micro-batch size to 8; for CoLA, we set the learning rate to 2.5e-6, learning rate warm-up steps to 250, max sequence length to 128, the macro-batch size to 32 and micro-batch size to 8. After the learning rate warm-up stage, we decay the learning rate linearly over the training epochs.

**Language Modeling.** For both WikiText and arXiv datasets, we use AdamW optimizer with a learning rate of 5.0e-6. We train the model for 10 epochs with a macro-batch size of 32 and a micro-batch size of 1. The max sequence length is set to 1024 for both datasets. After the learning rate warm-up stage, we decay the learning rate linearly over the training epochs.

## D  Distributed View of `AQ-SGD` lgorithm

Algorithm 2 shows a multi-node view of `AQ-SGD`. For brevity, we omit the first warm-up epoch, where we conduct uncompressed training, and thus we update the previous messages by $m(\xi) \leftarrow a(\xi, x)$.

---

**Algorithm 2** `AQ-SGD` Algorithm

---

**Initialize:** $x_0$, learning rate $\gamma$, network $a$'s weights $x^{(a)}$, network $b$'s weights $x^{(b)}$, quantization function $Q$, the arrays of previous messages $m$, where networks $a$ and $b$ each maintain a copy of it.

**for** t = 1, ..., T **do**

    (**on network** $a$)

    Randomly sample $\xi_t$

    $\Delta m(\xi_t) \leftarrow Q\big(a(\xi_t, x_t^{(a)}) - m(\xi_t)\big)$

    Update $m(\xi_t) \leftarrow m(\xi_t) + \Delta m(\xi_t)$

    Send $\Delta m(\xi_t)$ to network $b$

    (**on network** $b$)

    Update $m(\xi_t) \leftarrow m(\xi_t) + \Delta m(\xi_t)$

    Update $x_{t+1}^{(b)} \leftarrow x_t^{(b)} - \gamma \cdot \nabla_{x^{(b)}}(f \circ b)|_m$

    Send $Q(\nabla_a(f \circ b)|_m)$ to network $a$

    (**on network** $a$)

    Update $x_{t+1}^{(a)} \leftarrow x_t^{(a)} - \gamma \cdot Q(\nabla_a(f \circ b)|_m) \cdot \nabla_{x^{(a)}}a$

**end for**

**Output:** $x = (x_T^{(a)}, x_T^{(b)})$

---

# E  Decentralized Training over Slow Network

Decentralized training for large foundation models recently attracted intensive interests. Example projects include Learning@home [1], DeDLOC [2], and Training Transformers Together [3]. The goal of these projects is to enable a decentralized open-volunteering paradigm for foundation model training. As many geo-distributed users contribute their GPUs, these GPUs are often connected via slow networks. For example, [2] investigates a heterogeneous with bandwidths of 200Mbps, 100Mbps, and 50Mbps; [3] advocates to train Transformer over the Internet with 10-100Mbps bandwidth; [4] considers network bandwidth less than 400Mbps.

In this setting, communication compression is key to performance. However, when compressing activations, existing methods rely on direct quantization. This inspired our paper, which provides the first activation quantization method with rigorous theoretical guarantee and outperforms direct quantization.

# F  Discussion on Tensor Parallelism

We here discuss tensor parallelism [24], and the potential adaptation of our algorithm to it. In tensor parallelism, the activations are computed across different machines, and need to be aggregated. Therefore we need to compress activations both before and after allreduce to support tensor parallelism. Specifically, suppose $N$ machines conduct tensor parallelism, then the output activation is:

$$A = A_1 + A_2 + ... + A_N, \tag{F.1}$$

and we need to compress communication twice:

$$A_Q = Q[Q(A_1) + Q(A_2) + \ldots + Q(A_N)]. \tag{F.2}$$

We believe that delta compensation could be applied to all $Q(-)$, similar to how previous work handles gradient compression (e.g. Eq. 3 and 4 in [27]). However, this requires careful further studies, both empirically and theoretically. We leave activation compression for tensor parallelism as future work.

# G  Limitation and Potential Future Direction

**Additional Storage.**  Our algorithm trades storage for communication. Fortunately, we find this is a reasonable trade-off in our settings. In the following we show that we can offload the activations to SSD and hide it within the GPU computation of other data examples.

We compare the throughput under the bandwidth of 10Gb/s. FP32 achieves a throughput of 3.8 seqs/s, while `AQ-SGD`, either offloading activations to host memory or SSDs, achieves 4.0 seqs/s. Considering the similar training throughput of the above three settings, we show that the overhead of offloading to SSD can be successfully hided in GPU computation. In particular, our largest experiment in this paper requires 172GB storage per machine, which even with a 10x larger dataset can be easily offloaded to SSDs. For much larger datasets (e.g. 100x), we can use data parallelism to reduce the storage requirement for each machine. For an even larger dataset, our algorithm might not support it well. This then requires further studies.

**Online Learning.**  Our proposal relies on iterating over multiple epochs, which is a common setting. We understand our current algorithm has limitations in the settings such as online learning. In the following, we provide a potential solution (to both storage requirement and limitation in online learning) – relaxing `AQ-SGD` by clustering activations and storing only the centers of the clusters.

Recall that `AQ-SGD` compresses the "delta" (difference between activations from two epochs) and thereby needs to store activations from the previous epoch. In the relaxed `AQ-SGD`, we could use algorithms like clustering or locality sensitive hashing to partition the activations and then we only store the "centers" of each partition/cluster. When computing the "delta", we can first identify which partition/cluster the current activation belongs to and retrieve the corresponding "center". Then "delta" = activation - "center". This will potentially help address storage and online learning limitations. We will explore this in future work.

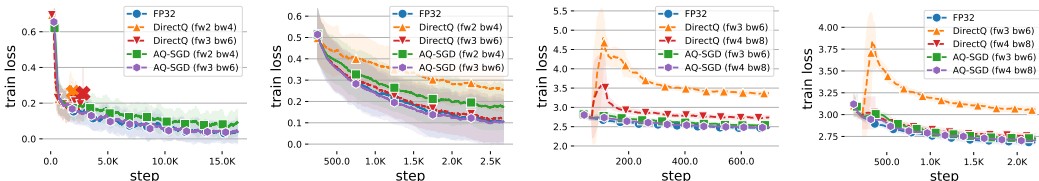

(a) QNLI, DeBERTa-1.5B  (b) CoLA, DeBERTa-1.5B  (c) WikiText2, GPT2-1.5B  (d) arXiv, GPT2-1.5B

Figure 6: Convergence (loss vs. # stpes) of different approaches. $\times$ represents divergence.

Table 5: Training Throughput.

| Network Bandwidth | DeBERTa-1.5B, QNLI | | | GPT2-1.5B, WikiText2 | | |
|---|---|---|---|---|---|---|
| | FP32 | DirectQ fw2 bw4 / fw3 bw6 | AQ-SGD fw2 bw4 / fw3 bw6 | FP32 | DirectQ fw3 bw6 / fw4 bw8 | AQ-SGD fw3 bw6 / fw4 bw8 |
| 10 Gbps | $12.9_{\pm 0.02}$ | $13.6_{\pm 0.02}$ / $13.6_{\pm 0.02}$ | $13.6_{\pm 0.02}$ / $13.5_{\pm 0.02}$ | $3.8_{\pm 0.01}$ | $4.0_{\pm 0.01}$ / $4.1_{\pm 0.01}$ | $4.0_{\pm 0.01}$ / $4.0_{\pm 0.01}$ |
| 1 Gbps | $9.6_{\pm 0.02}$ | $13.3_{\pm 0.02}$ / $13.1_{\pm 0.02}$ | $13.3_{\pm 0.02}$ / $13.0_{\pm 0.02}$ | $3.2_{\pm 0.01}$ | $4.0_{\pm 0.01}$ / $4.0_{\pm 0.01}$ | $4.0_{\pm 0.01}$ / $3.9_{\pm 0.01}$ |
| 500 Mbps | $6.2_{\pm 0.03}$ | $13.0_{\pm 0.03}$ / $12.6_{\pm 0.03}$ | $12.9_{\pm 0.03}$ / $12.5_{\pm 0.03}$ | $2.7_{\pm 0.02}$ | $3.9_{\pm 0.01}$ / $3.9_{\pm 0.01}$ | $3.9_{\pm 0.01}$ / $3.9_{\pm 0.01}$ |
| 300 Mbps | $4.4_{\pm 0.04}$ | $12.5_{\pm 0.02}$ / $11.9_{\pm 0.03}$ | $12.4_{\pm 0.03}$ / $11.8_{\pm 0.03}$ | $1.8_{\pm 0.02}$ | $3.9_{\pm 0.01}$ / $3.8_{\pm 0.01}$ | $3.8_{\pm 0.01}$ / $3.8_{\pm 0.01}$ |
| 100 Mbps | $1.6_{\pm 0.04}$ | $10.7_{\pm 0.03}$ / $9.4_{\pm 0.03}$ | $10.6_{\pm 0.03}$ / $9.1_{\pm 0.03}$ | $0.5_{\pm 0.02}$ | $3.5_{\pm 0.02}$ / $3.0_{\pm 0.02}$ | $3.4_{\pm 0.01}$ / $3.0_{\pm 0.02}$ |

| Network Bandwidth | DeBERTa-1.5B, CoLA | | | GPT2-1.5B, arXiv | | |
|---|---|---|---|---|---|---|
| | FP32 | DirectQ fw2 bw4 / fw3 bw6 | AQ-SGD fw2 bw4 / fw3 bw6 | FP32 | DirectQ fw3 bw6 / fw4 bw8 | AQ-SGD fw3 bw6 / fw4 bw8 |
| 10 Gbps | $17.1_{\pm 0.03}$ | $18.0_{\pm 0.03}$ / $17.9_{\pm 0.03}$ | $17.9_{\pm 0.03}$ / $17.8_{\pm 0.03}$ | $3.8_{\pm 0.01}$ | $4.0_{\pm 0.01}$ / $4.1_{\pm 0.01}$ | $4.0_{\pm 0.01}$ / $4.0_{\pm 0.01}$ |
| 1 Gbps | $12.2_{\pm 0.03}$ | $17.4_{\pm 0.02}$ / $17.1_{\pm 0.02}$ | $17.3_{\pm 0.02}$ / $16.9_{\pm 0.02}$ | $3.2_{\pm 0.01}$ | $4.0_{\pm 0.01}$ / $4.0_{\pm 0.01}$ | $4.0_{\pm 0.01}$ / $3.9_{\pm 0.01}$ |
| 500 Mbps | $8.9_{\pm 0.03}$ | $16.7_{\pm 0.03}$ / $16.2_{\pm 0.03}$ | $16.7_{\pm 0.03}$ / $16.1_{\pm 0.03}$ | $2.7_{\pm 0.02}$ | $3.9_{\pm 0.01}$ / $3.9_{\pm 0.01}$ | $3.9_{\pm 0.01}$ / $3.9_{\pm 0.01}$ |
| 300 Mbps | $6.0_{\pm 0.04}$ | $16.1_{\pm 0.03}$ / $15.2_{\pm 0.03}$ | $16.0_{\pm 0.03}$ / $15.1_{\pm 0.03}$ | $1.8_{\pm 0.02}$ | $3.9_{\pm 0.01}$ / $3.8_{\pm 0.01}$ | $3.8_{\pm 0.01}$ / $3.8_{\pm 0.01}$ |
| 100 Mbps | $2.2_{\pm 0.04}$ | $13.1_{\pm 0.03}$ / $11.5_{\pm 0.03}$ | $13.1_{\pm 0.03}$ / $11.3_{\pm 0.03}$ | $0.5_{\pm 0.03}$ | $3.5_{\pm 0.01}$ / $3.0_{\pm 0.02}$ | $3.4_{\pm 0.01}$ / $3.0_{\pm 0.01}$ |

# H  Additional Results

We provide additional experimental results. Specifically, we show:

- the convergence results with standard deviation;
- the training throughput for different dataset settings;
- the numerical stability of training from scratch;
- the training results under FP16 precision;
- the robustness of `AQ-SGD` under different hyperparameter settings;
- the effectiveness of `AQ-SGD` in the split learning scenario.

## H.1  Convergence Results with Standard Deviation

In the main content, we show the convergence performance of different approaches. We repeated each experiment three times to ensure reproducibility. We calculate the moving averages of these convergence curves and then average the results of repeated experiments. We visualize (shaded areas) the moving standard deviation in all repeated experiments in Figure 6. Overall, we observe consistent results for all datasets.

## H.2  Throughput under Different Dataset Settings

We show the training throughput under different dataset settings in Figure 5. In general, the observation is similar to that of the main content: our approach maintains similar throughput even when the network is $100\times$ slower (from 10Gbps to 100Mbps). WikiText2 and arXiv have essentially the same throughput results, since we use the same training settings for them.

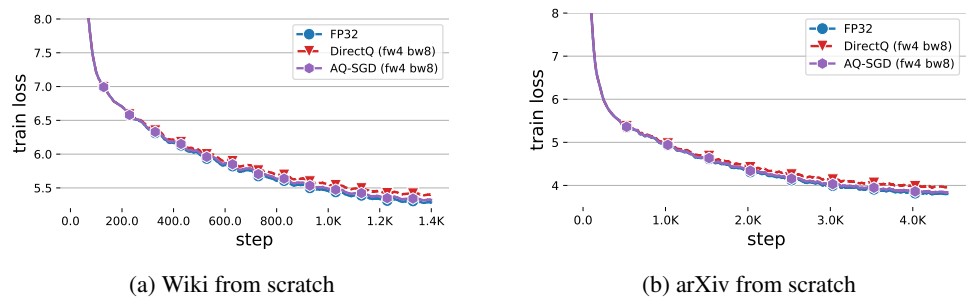

(a) Wiki from scratch            (b) arXiv from scratch

Figure 7: Convergence results of training from scratch.

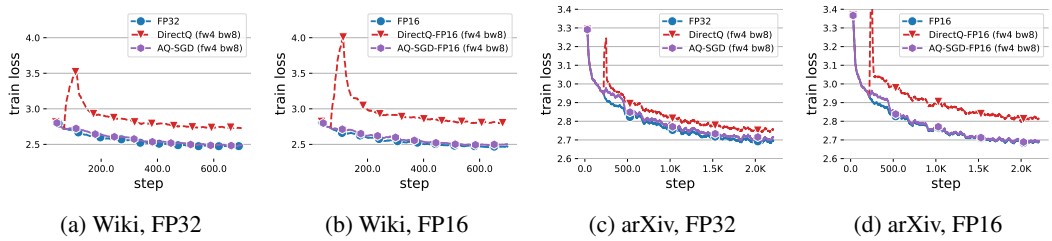

(a) Wiki, FP32     (b) Wiki, FP16     (c) arXiv, FP32     (d) arXiv, FP16

Figure 8: Comparison of fine-tuning under FP32 and FP16.

### H.3 Results of Training from Scratch

We investigate the numerical stability of `AQ-SGD` by showing the convergence result of training from scratch, where the model parameters are randomly initialized. We train WikiText and arXiv datasets for 20 epochs and use the first 10% of steps as warm-up, respectively. As shown in Figure H.3, we can see that `AQ-SGD` converges almost as fast as FP32 when training from scratch, which indicates our approach is robust enough even when the model is far from the converged state. In contrast, the curve of `DirectQ` becomes flatter in the late training stage, showing a clear gap with FP32.

### H.4 Results of Training under FP16

To investigate the convergence performance of `AQ-SGD` under low-precision training, we here show the results of FP16 training, and compare it with FP32 training. Figure 8 compares the results under FP32 and FP16. In general, the convergence curves are consistent with the FP32 case. This confirms the effectiveness of `AQ-SGD` when the activation is already in low precision.

### H.5 Hyper-parameter Sensitivity

Here we demonstrate the robustness of our method in various settings. For fast validation, we focus on evaluating DeBERTa-v3-base[6] on QNIL and CoLA datasets. We by default use $K = 4$ devices for pipeline parallel training, 2 bits for forward activation, and 4 bits for backward gradients (`fw2 bw4`).

**Number of Pipeline Stages.** We first investigate the influence of the number of pipeline stages on convergence performance. Intuitively, partitioning into more pipeline stages leads to more rounds of data compression and communication,resulting in a larger accumulated compression error. The results of Figures 9a and 9b confirm this intuition. Specifically, the direct quantization method works not bad when $K = 2$, but its performance becomes unsatisfied when we further enlarge $K$. In comparison, our approach can maintain similar convergence performance to FP32.

**Number of Bits in Communication.** Figures 9c and 9d compare different methods with different numbers of bits in communication. We observe that using more bits can improve the convergence performance but lead to higher communication overheads. In general, our approach achieves better accuracy-efficiency trade-offs.

---

[6]https://huggingface.co/microsoft/deberta-v3-base

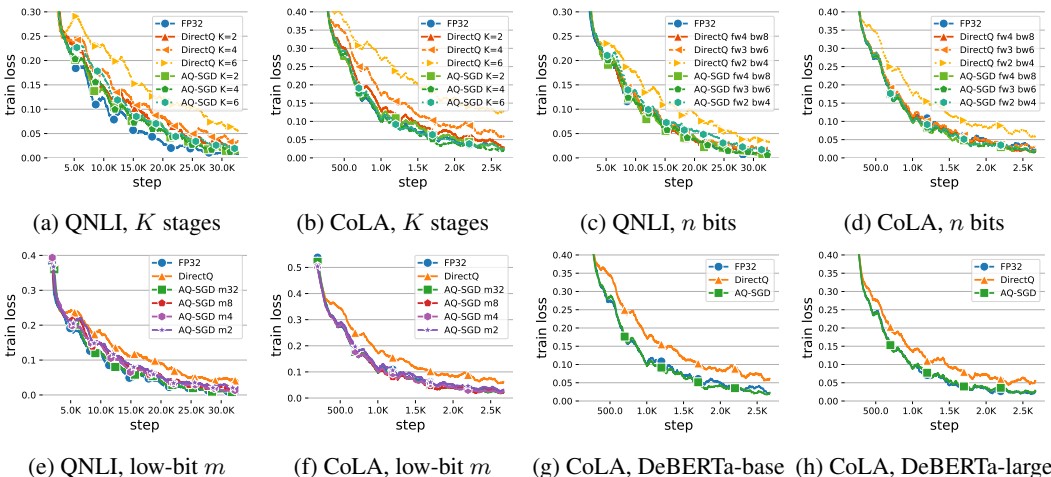

| (a) QNLI, $K$ stages | (b) CoLA, $K$ stages | (c) QNLI, $n$ bits | (d) CoLA, $n$ bits |

| (e) QNLI, low-bit $m$ | (f) CoLA, low-bit $m$ | (g) CoLA, DeBERTa-base | (h) CoLA, DeBERTa-large |

Figure 9: Convergence curves under different configurations.

**Number of Bits for Previous Messages.** We may find that storing all previous messages is space-intensive. To reduce such requirements, we show that previous messages $m$ can be preserved with low precision. We here perform quantization on $m$, where $\texttt{m}z$ means that we use $z$ bits for previous messages. Figures 9e and 9f show the results with different number of bits of the previous messages. When only 2 bits are used for the previous messages, despite the fact that it is slightly worse than our default setting, our approach is still significantly better than `DirectQ`. And there is no significant performance drop when 8 bits are used for the previous messages.

**Pre-trained Model Sizes.** Figures 9g and 9h show the results of the base and large version of DeBERTa. Surprisingly, larger models seem to be more tolerant of errors from activation compression than smaller models. One possible reason is that larger models usually use much smaller learning rates. So the error of each iteration can be restricted to a smaller range. Here, we use 2.0e-5 for the base model and 7e-6 for the large model, as suggested in the official repository of DeBERTa.

### H.6 Split Learning

Split learning is a scenario of federated learning, where the client trains a shallow part of deep network (known as the cut layer) that accesses the training data, while the rest of the model is trained in a data center. Clients and server need to exchange the activation and its gradients in the cut, where `AQ-SGD` can be adopted. We evaluated `AQ-SGD` on a split learning scenario where neither the input data nor its labels are shared with the server—the model is cut twice, one after the first resnet block and one before the last block to generate the prediction. We evaluate `AQ-SGD` for split learning over Cifar10 and Cifar100 with the ResNet34 model. We set 16 clients and use a Dirichlet distribution with concentration parameter 0.5 to synthesize non-identical datasets. Following the previously established work of split learning, in each communication round, we conduct local training for each client sequentially. and each client will train 3 epochs with its local data. We utilize SGD optimizer with momentum of 0.9, a batch size of 64, and a learning rate of 0.01. We decay the learning rate to its 10% for every 20 communication rounds.

The datasets are augmented with random cropping and flipping. To adapt to random cropping, we do the same cropping operation on the retrieved previous message, and only update its non-cropped part. To adapt to random flipping, we maintain another previous message copy for flipped images, and retrieve and update only the corresponding copy during training.

Figure 10 presents the results of split learning, where `fw2 bw8[0.2]` means that, for forward pass, we perform 2-bit quantization, and for backward pass, we keep only the top 20% gradients and then perform 8-bit quantization. We can see that `AQ-SGD` transfers the activations in 2 bits while maintaining a performance similar to `FP32`, which indicates the effectiveness of `AQ-SGD` in improving the communication efficiency in the split learning scenario. Furthermore, compared to `DirectQ`, `AQ-SGD` shows advantages in terms of both the convergence and generalization of the trained model.

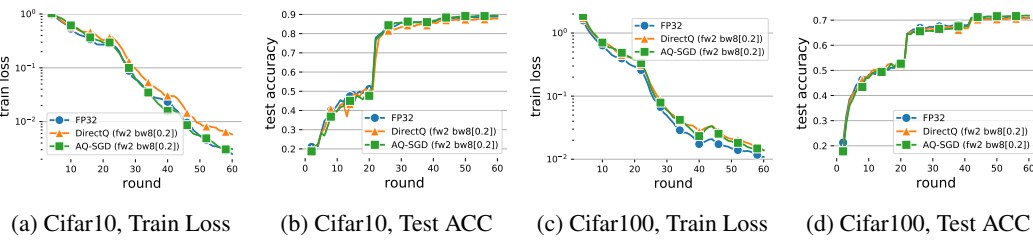

| (a) Cifar10, Train Loss | (b) Cifar10, Test ACC | (c) Cifar100, Train Loss | (d) Cifar100, Test ACC |

Figure 10: Results of split learning with ResNet34.

# I  Case Study of Generation Results

We here conduct case study to better understand the generation quality across different methods. All methods are fine-tuned on WikiText with the same seed. We use greedy decoding to generate texts. For `DirectQ` and `AQ-SGD`, we use 4-bit in forward pass and 8-bit in backward pass.

We randomly pick documents from the test set of WikiText to construct the prompts and let the model complete the rest. The generation results after fine-tuning are presented in Tables 6 and 7. For instance, in Cases 1 and 2, `AQ-SGD` produces the same results as `FP32`; in Case 3, although `AQ-SGD` does not generate the exact same output as `FP32`, it still expresses a similar meaning. This indicates our proposed compression scheme has relatively little impact on the learning process. In contrast, `DirectQ` may generate text that is not relevant to the context. For example, in Case 1, we know that "Matsuo Bashō was strongly influenced by Du Fu", so we should expect the text below to show the fact how he was influenced, just like `FP32` and `AQ-SGD` did. But `DirectQ` fails to generate reasonable text based on the context. Other inference results presented in Tables 6 and 7 lead to the same observation.

Table 6: Case study of generation results.

| **Case 1** |
| --- |

| | |
| --- | --- |
| **Prompt:** | ... Matsuo Bashō, the greatest haiku poet, was also strongly influenced by Du Fu; in Oku no Hosomichi, his masterpiece, he cites the first two lines of A Spring View before a haiku as its introduction and also many of his other haiku have similar wording and themes. It is said that when he died |
| **Gold:** | in Osaka during a long travel, a copy of Du Fu's poetry was found with him as one of a few precious items which he was able to carry around. |
| FP32: | , his body was placed in a coffin with a Du Fu-like inscription. |
| AQ-SGD: | , his body was placed in a coffin with a Du Fu-like inscription. |
| DirectQ: | , his body was covered with a red cloth and his head was shaved. |

| **Case 2** |
| --- |

| | |
| --- | --- |
| **Prompt**: | ... Peppiatt notes that the exhibition showed Bacon no longer needed sensationalist material to make an impact, and was now |
| **Gold**: | capable of creating an intense emotional response through more subtle means ... |
| FP32: | able to "make a serious contribution to the art of painting" |
| AQ-SGD: | able to "make a serious contribution to the art of painting" |
| DirectQ: | able to "make a serious statement about the state of the art" |

| **Case 3** |
| --- |

| | |
| --- | --- |
| **Prompt**: | ... At this point, Dokhturov's column appeared behind the French line and joined the battle. The French were outnumbered more than three to |
| **Gold**: | one, assaulted in the front by Miloradovich's column, in the middle by Strik's and in the rear by Dokhturov. |
| FP32: | one, but they held their ground and forced the Russians to retreat. |
| AQ-SGD: | one, but they were able to hold their ground. |
| DirectQ: | one, and the Russians were able to hold their ground. |

| **Case 4** |
| --- |

| | |
| --- | --- |
| **Prompt**: | ... Between Jupiter and Fort Pierce, the storm knocked down power and telegraph lines. In the latter city, high waves washed out a portion of the causeway. In the 1980s, an elderly resident recalled that the storm was the |
| **Gold**: | most severe on record in Fort Pierce. |
| FP32: | worst storm he had ever seen. |
| AQ-SGD: | worst storm he had ever seen. |
| DirectQ: | worst he had ever experienced. |

| **Case 5** |
| --- |

| | |
| --- | --- |
| **Prompt**: | ... Lesnar was born in Webster, South Dakota, on July 12, 1977. He was raised on a Webster dairy farm owned by his parents, Stephanie and Richard Lesnar. He is of German descent. He has two older brothers named Troy and Chad, and a younger sister named Brandi. At age 17, he joined the National Guard, where he |
| **Gold**: | was assigned to an office job after his red-green colorblindness was deemed hazardous to his desire to work with explosives. |
| FP32: | served in the South Dakota National Guard. He was a member of the South Dakota National Guard for two years. |
| AQ-SGD: | served in the South Dakota National Guard. He was a member of the South Dakota National Guard for two years. |
| DirectQ: | served in the South Dakota National Guard for two years. He was discharged in May 1991. |

Table 7: Case study of generation results (more).

| Case 6 | |
|---|---|
| **Prompt**: | ... Mega placed the game at # 7 in their "Top Mega Drive Games of All Time" list. MegaTech magazine praised the game's graphics, but criticized the game's easy difficulty level, |
| **Gold**: | explaining simply that "the graphics are excellent, but the game is easy to complete." |
| `FP32`: | stating that "the game is too easy to be enjoyable". |
| `AQ-SGD`: | stating that it was "too easy". |
| `DirectQ`: | which was criticized for being too easy. |

| Case 7 | |
|---|---|
| **Prompt**: | ... "Obviously, he has great rapport and communication ability with his team." Yahoo! Sports compared Stevens to legendary coach John Wooden writing " |
| **Gold**: | Brad Stevens is winning at Butler the Wooden way – calm and composed on the sideline." |
| `FP32`: | He's a coach who has been around the block, and he's not afraid to tell you what he thinks." |
| `AQ-SGD`: | Stevens is a coach who has been around the game for a long time, and he has a knack for getting the most out of his players." |
| `DirectQ`: | Stevens is a coach who has been able to mold a team into a championship contender. He has a knack for getting the most out of his players, and he has a knack for getting the most out of his players' personalities." |

| Case 8 | |
|---|---|
| **Prompt**: | ... The Gore's social circle in England revolved around his good friend Rufus King, who |
| **Gold**: | was appointed Ambassador to Great Britain in 1796, along with other Massachusetts expatriates. |
| `FP32`: | was a member of the Royal Society. |
| `AQ-SGD`: | was a member of the Royal Society. |
| `DirectQ`: | was a member of the Royal Society. |

| Case 9 | |
|---|---|
| **Prompt**: | ... The city has 896 barangays that are known by sequential numbers instead of names. These barangays are further group into |
| **Gold**: | 100 zones for administrative and municipal purposes. |
| `FP32`: | districts, which are further subdivided into wards. |
| `AQ-SGD`: | districts, which are further subdivided into wards. |
| `DirectQ`: | districts, which are numbered according to the number of barangays in the district. |

| Case 10 | |
|---|---|
| **Prompt**: | ... The National Spiritual Assembly of the Bahá 'ís of the Philippines, the governing body of the Filipino Bahá'í community, is |
| **Gold**: | headquartered near Manila's eastern border with Makati. |
| `FP32`: | headquartered in the city. |
| `AQ-SGD`: | headquartered in the city. |
| `DirectQ`: | headquartered in the city. |