# OpenReview forum: "Fine-tuning Language Models over Slow Networks using Activation Quantization with Guarantees"
_NeurIPS.cc/2022/Conference — NeurIPS 2022 Accept_

### Official Review · Reviewer_dFk7 · 2022-07-10

**Rating:** 6
**Confidence:** 3
**Soundness:** 3 good
**Presentation:** 3 good
**Contribution:** 3 good

**Summary:**

Large Language models are trained or finetuned using a combination of pipeline and data parallelism methods. Pipeline parallelism leads to activation and activation gradient being communicated between devices, while data parallelism leads to weight gradients being communicated between devices. A lot of work has focussed on the later, but the former has received less focus. Compressing activations is non trivial as compressing it in a stochastic unbiased way will lead to biases in the gradient  and breaks the unbiasedness assumption made by most gradient compression results. Previous work either failed to provide theoretical guarantees

This paper proposes AC-SGD, an activation compression algorithm for communication efficient pipeline parallelism. This method helps accelerate pipeline parallelism over slow networks. They provide theoretical guarantees to prove convergence and show that the method can be implemented with minimum runtime overhead (albeit at a huge memory cost). They compare their method with recent work and show better accuracy at lower precision. Depending on the communication BW used as baseline, they can achieve 4.3x end to end speedup.

**Questions:**

Questions -

1. What was the storage requirement for each of the benchmarks evaluated in the paper?
2. What motivated the path to explore compressing the delta of activation per sample rather than the activation itself


**Ethics Review Area:**

["I don’t know"]

**Limitations:**

Have the authors adequately addressed the limitations and potential negative societal impact of their work? Yes

**Strengths And Weaknesses:**

Strengths:

1. The claims in the paper are evaluated against strong baselines
2. The method comes with a theoretical guarantee of convergence

Weakness:

1. The idea relies on data being repeated over multiple epochs. This can limit the applicability of this techniques to more important workloads like pre-training and limit to the idea to only fine-tuning of LLM. This reduces the potential impact of the work.

2. The requirement to store activations for new samples in the finetuning datasets makes it harder to scale this technique with larger finetuning datasets

---

> ### Author Response · Authors · 2022-08-02
> **Response to Reviewer (1/1)**
>
> Thank you for the comments. Our responses to your questions can be found as follows:
>
> **Q1: The idea relies on data being repeated over multiple epochs. This can limit the applicability of this techniques to more important workloads like pre-training and limit to the idea to only fine-tuning of LLM. This reduces the potential impact of the work. The requirement to store activations for new samples in the finetuning datasets makes it harder to scale this technique with larger finetuning datasets**
>
> **Response:** We appreciate the comment. We here first discuss our storage requirement and limitation in online learning. Then we provide a potential solution to address both for the future work. We added a section in Appendix G, discussing the limitations and potential future directions.
>
> Storage requirement: Indeed, our algorithm trades storage for communication. Fortunately, we find this is a reasonable trade-off in our settings – in the following we show that we can offload the activations to SSD and hide it within the GPU computation of other data examples.
>
> By comparing the throughput (# seqs/second) under the bandwidth of 10Gb/s (slower network will have even smaller gap), we find that the three cases depicted below have similar training throughput.
>
>
> | FP32 | AQ-SGD offloading to host memory | AQ-SGD offloading to SSD |
> | :--- | :------------------------------- | :----------------------- |
> | 3.8  | 4.0                              | 4.0                      |
>
> Therefore:
>
> - Currently, our largest experiment requires <=172GB storage per machine; even with a 10x larger dataset, this can be easily offloaded to SSDs.
> - For much larger datasets (e.g. x100), we can use data parallelism to reduce the storage requirement for each machine;
> - For an even larger dataset, our algorithm might not support it well. This requires further studies.
>
> Online learning: Our proposal indeed relies on iterating over multiple epochs. Although it is a quite common setting, we do agree that our current algorithm has limitations in settings such as online learning. In the following, we provide a potential solution – relaxing AQ-SGD by clustering activations and storing only the centers of the clusters. We will definitely explore this exciting idea as future work.
>
> Potential solution to both storage requirement and limitation in online learning: Recall that AQ-SGD compresses the "delta" (difference between activations from two epochs) and thereby needs to store activations from the previous epoch. In the relaxed AQ-SGD, we could use algorithms like clustering or locality sensitive hashing to partition the activations and then we only store the "centers" of each partition/cluster. When computing the "delta", we can first identify which partition/cluster the current activation belongs to and retrieve the corresponding "center". Then "delta" = activation - "center". This could potentially help address storage and online learning limitations.
>
> **Q2: What was the storage requirement for each of the benchmarks evaluated in the paper?**
>
> **Response:** The storage requirements for each of the benchmarks evaluated in the paper are as follows: QNLI: 172 GB per node; CoLA: 14 GB per node; Wikitext2: 15 GB per node; Arxiv: 50 GB per node.
>
> In our response to Q1, we show that we can offload the activations to SSD and hide it within the GPU computation of other data examples. So using an SSD will not bring any additional training time. Currently, our largest experiment requires 172GB storage per machine; and even with a 10x larger dataset, this can be easily offloaded to SSD.
>
> **Q3: What motivated the path to explore compressing the delta of activation per sample rather than the activation itself?**
>
> **Response:** Thanks for your question! First, directly compressing activations in a stochastically unbiased way will lead to biases in the gradient that cannot be measured easily or expressed in closed form, which breaks the unbiasedness assumptions adopted in previous work. Also, we empirically find that performing direct quantization over activations leads to poor convergence performance. Meanwhile, we observe that the activation changes at a small pace during fine-tuning, and the delta of activation becomes smaller as training converges. This inspires us to compress and communicate the delta rather than the activation itself.

---

### Official Review · Reviewer_uCZr · 2022-07-11

**Rating:** 8
**Confidence:** 4
**Soundness:** 3 good
**Presentation:** 3 good
**Contribution:** 3 good

**Summary:**

This work examines the feasibility to compress the activations for models trained with pipeline parallelism. To this end, the scheme AC-SGD is proposed, which aims to make pipeline-parallel training more communication-efficient over slot networks. Different from previous efforts in activation compression, instead of compressing activation values directly, AC-SGD compresses the changes of the activations. The most novel insight is that: one can still achieve O(1/pT) convergence rate for non-convex objectives under  activation compression, without making assumptions on gradient unbiasedness that  do not hold for deep learning models with non-linear activation functions. The this work shows that AC-SGD can be optimized and implemented efficiently, without additional end-to-end runtime overhead. The evaluations on AC-SGD are shown on fine-tuned language models with up to 1.5 billion parameters, compressing activations to 2-4 bits. AC-SGD provides up to 4.3⇥ end-to-end speed-up in slower networks, without sacrificing model quality.

**Questions:**

- Can you provide more evidence to back up your simulation setups on the slow work?
- How your optimizations benefit the overall system (separately and collectively)?

**Limitations:**

- I think the paper has good potentials, and it can be strengthened by addressing the two points (raised in the weakness part).
- I assume the geo-distributed network is a more realistic scenario for this proposed approach. Hence, I would suggest the authors to get more literature on this part, and back up both their motivation and simulation setup better (both qualitatively and quantitatively).
- Also, I believe breaking down the optimizations in AC-SGD can add more values in this work, so that the readers can better understand how essential your optimizations are.


**Strengths And Weaknesses:**

Strengths:
- An interesting insight on pipeline-parallel training.
- The solution is effective and the optimization space is covered.
- Extensive evaluation results.
- Source codes available

Weaknesses:
- The work may need more rationales upfront to motivate the problems (i.e. slow networks)
- More breakdown analysis to showcase the effectiveness of the approach

---

> ### Author Response · Authors · 2022-08-02
> **Response to Reviewer (1/1)**
>
>
> We thank you for the comments and suggestions! Our responses to your questions are as follows:
>
> **Q1: The work may need more rationales upfront to motivate the problems (i.e. slow networks). I assume the geo-distributed network is a more realistic scenario for this proposed approach. Hence, I would suggest the authors to get more literature on this part, and back up both their motivation and simulation setup better (both qualitatively and quantitatively).**
>
> **Response:** We agree with the suggestion! We have added more motivation and details about slow network setting (our target application scenarios) both in Introduction and Appendix E.
>
> Our target application is decentralized training for large foundation models, a scenario that recently attracted intensive interests. Example projects include Learning@home [6], DeDLOC [7], and Training Transformers Together [2]. The goal of these projects is to enable a decentralized open-volunteering paradigm for foundation model training. As many geo-distributed users contribute their GPUs, these GPUs are connected via slow networks. For example, [1] investigates a heterogeneous with bandwidths of 200 Mb/s, 100 Mb/s, and 50 Mb/s; [2] advocates to train Transformer over the Internet with 10-100 Mb/s bandwidth; [3] considers network bandwidth less than 400Mb/s.
>
> In this setting, communication compression is key to performance. However, existing methods rely on direct quantization of activations. This inspired our paper, which provides the first activation compression method with rigorous theoretical guarantee and outperforms direct quantization.
>
>
>
> [1] Diskin, Michael, et al. "Distributed deep learning in open collaborations." Advances in Neural Information Processing Systems 34 (2021): 7879-7897.
>
> [2] Borzunov, Alexander, et al. "Training Transformers Together." NeurIPS 2021 Competitions and Demonstrations Track. PMLR, 2022.
>
> [3] Ryabinin, M., Dettmers, T., Diskin, M., & Borzunov, A. (2021). SWARM Parallelism: Training Large Models Can Be Surprisingly Communication-Efficient.
>
> [4] Yuan, Binhang, et al. "Decentralized Training of Foundation Models in Heterogeneous Environments." arXiv preprint arXiv:2206.01288 (2022).
>
> [5] Beberg, Adam L., et al. "Folding@ home: Lessons from eight years of volunteer distributed computing." 2009 IEEE International Symposium on Parallel & Distributed Processing. IEEE, 2009.
>
> [6] Ryabinin, Max, and Anton Gusev. "Towards crowdsourced training of large neural networks using decentralized mixture-of-experts." Advances in Neural Information Processing Systems 33 (2020): 3659-3672.
>
>
>
> **Q2:** **I believe breaking down the optimizations in AC-SGD can add more values in this work, so that the readers can better understand how essential your optimizations are.**
>
> **Response:** Thank you for the suggestion! In the paper, we try to compress all data exchanges between machines, including model gradients, forward activations, and backward gradients. To understand their benefit separately and collectively, we show in Figure 5(c) that both activation and gradient compression are important in terms of improving end-to-end training throughput, and disabling any of them will lead to a much lower training throughput.
>
> Furthermore, we here show the breakdown of AQ-SGD (fw4 bw8) per micro batch for GPT2-1.5B:
>
> | Bandwidth | Fw comp. | Bw comp. | Fw comm. | Bw comm. |
> |-----|-----|-----|-----|-----|
> | 500 Mbps  | 45ms     | 135ms    | 13ms     | 25ms     |
> | 200 Mbps  | 45ms     | 135ms    | 31ms     | 63ms     |
> | 100 Mbps  | 45ms     | 135ms    | 63ms     | 125ms    |
>
> We note that computation and communication can overlap, so the actual time is the larger one of the two.
>
> If the reviewer wants to see other breakdowns and analysis, please let us know and we will present them in the discussion phase.

---

> > ### Comment · Reviewer_uCZr · 2022-08-03
> > **Score Raised after Rebuttal & Revision**
> >
> > Thanks for your careful rebuttal and revision, which I highly appreciate. After reading the rebuttal and revision, I decide to raise the score (to 8) and the breakdown analysis may help with the further discussions (and score adjustment), if the authors can make them available. I assume this paper has done a very great job, and it deserves the acceptance (from my perspective).

---

> > > ### Author Response · Authors · 2022-08-04
> > > **Thank you for your appreciation!**
> > >
> > > Thank you so much for your appreciation and feedback! We have now integrated the breakdown in Table 3 in the main paper.
> > > Please feel free to let us know if there is anything else we can do to improve the paper!

---

### Official Review · Reviewer_XUdm · 2022-07-12

**Rating:** 5
**Confidence:** 5
**Soundness:** 2 fair
**Presentation:** 3 good
**Contribution:** 2 fair

**Summary:**

The authors proposed a differential activation quantization scheme with proven convergence, for distributed model-parallel training.


**Questions:**

- A biggest limitation of differential compression is the necessity for storage of the entire epoch of activations, and second biggest, the uselessness in online learning scenarios where data never repeat.
- Numerical stability probably needs more validation.  Though more stable than DirectQ, AC-SGD did present non-monotonic training loss, e.g. Fig. 5b.  Given that these are fine-tuning of pre-trained LMs, which are known to be already close in parameter space to convergence, it is questionable whether AC-SGD was stable at initialization for training from scratch.
- Following the same thought, as the authors mentioned in the manuscript, near-convergence training dynamics is probably more tolerant to aggressive activation quantization.
- Though SGD convergence proof is in the order, numerically it is usually more complicated in practice when diverse regularizers and different optimizers were used.  Is there any commonly used regularization or optimizers that do not agree with AC-SGD?
- What is the $Q(\cdot)$?  Does it involve affine calibration from the differential data statistics?  If so, what's the procedure?
- What if activation is already in low precision, say in FP16 in mixed-precision training?  Is AC-SGD still better off?  What about FP8?
- "Activation quantization" is perhaps more explicit and accurate than "activation compression".


**Limitations:**

See above.

**Strengths And Weaknesses:**

[+] Clear presentation in general

[+] Real-world demonstration

[-] Limitation of practical usefulness from storage requirement

[-] Lack of stability study

---

> ### Author Response · Authors · 2022-08-02
> **Response to Reviewer (2/2)**
>
>
> **Q4: What is the Q()? Does it involve affine calibration from the differential data statistics? If so, what's the procedure?**
>
> **Response**: We do a normalization per micro batch, and send the scaling information together with the quantized data (More details in Section 4.1)
>
>
>
> **Q5: What if activation is already in low precision, say in FP16 in mixed-precision training? Is AC-SGD still better off? What about FP8?**
>
> **Response:** Thanks for the suggestion! We now add a new experiment under FP16 precision. The results are now presented in the revised Appendix H.4. In general, the convergence curves are consistent with the FP32 case. This shows the robustness of AQ-SGD when the activation is already in low precision.
>
> For FP8, we have not found previous work about FP8 training especially for training large-scale foundation models. If the reviewer has papers in mind, let us know and we will definitely run it.
>
>
> **Q6: "Activation quantization" is perhaps more explicit and accurate than "activation compression".**
>
> **Response:** Thank you for your suggestion! We now changed “activation compression” to “activation quantization” in the revision.

---

> ### Author Response · Authors · 2022-08-02
> **Response to Reviewer (1/2)**
>
> Thanks for your suggestions, they have helped us to improve the paper!
>
> **Q1: A biggest limitation of differential compression is the necessity for storage of the entire epoch of activations, and second biggest, the uselessness in online learning scenarios where data never repeat.**
>
> **Response:** We first discuss our storage requirement and limitation in online learning. Then we provide a potential solution to address both for the future work. We  added a section in Appendix G, discussing the limitations and potential future directions.
>
> Storage requirement: Indeed, our algorithm trades storage for communication. Fortunately, we find this is a reasonable trade-off in our settings – in the following we show that we can offload the activations to SSD and hide it within the GPU computation of other data examples.
>
> By comparing the throughput (# seqs/second) under the bandwidth of 10Gb/s (slower network will have even smaller gap), we find that the three cases depicted below have similar training throughput.
>
> | FP32 | AQ-SGD offloading to host memory | AQ-SGD offloading to SSD |
> | ---- | -------------------------------- | ------------------------ |
> | 3.8  | 4.0                              | 4.0                      |
>
> Therefore:
> -  Currently, our largest experiment requires <=172GB storage per machine; even with a 10x larger dataset, this can be easily offloaded to SSDs.
> - For much larger datasets (e.g. 100x larger), we can use data parallelism to reduce the storage requirement for each machine;
> - For an even larger dataset, our algorithm might not support it well. This requires further studies.
>
> Online learning: Our proposal indeed relies on iterating over multiple epochs. Although it is a quite common setting, we do agree that our current algorithm has limitations in settings such as online learning. In the following, we provide a potential solution – relaxing AQ-SGD by clustering activations and storing only the centers of the clusters. We will definitely explore this exciting idea as future work.
>
> Potential solution to both storage requirement and limitation in online learning: Recall that AQ-SGD compresses the "delta" (difference between activations from two epochs) and thereby needs to store activations from the previous epoch. In the relaxed AQ-SGD, we could use algorithms like clustering or locality sensitive hashing to partition the activations and then we only store the "centers" of each partition/cluster. When computing the "delta", we can first identify which partition/cluster the current activation belongs to and retrieve the corresponding "center". Then "delta" = activation - "center". This could potentially help address storage and online learning limitations.
>
>
>
> **Q2: Numerical stability probably needs more validation. Though more stable than DirectQ, AC-SGD did present non-monotonic training loss, e.g. Fig. 5b. Given that these are fine-tuning of pre-trained LMs, which are known to be already close in parameter space to convergence, it is questionable whether AC-SGD was stable at initialization for training from scratch.   Following the same thought, as the authors mentioned in the manuscript, near-convergence training dynamics is probably more tolerant to aggressive activation quantization.**
>
> **Response:** We thank the reviewer for the comment. We add new experiments by training wikitext and arxiv for 20 epochs from scratch (with 10% as warm-up, same as our fine-tuning setting). The results can be found in the updated Appendix H.3. The results show that AQ-SGD converges almost as fast as FP32 when training from scratch, which indicates the numerical stability of our proposed approach. In contrast, the loss curve of DirectQ becomes flatter in the late training stage, showing a clear gap to FP32.
>
> | Loss     | FP32 | DirectQ (fw4 bw8) | AQ-SGD (fw4 bw8) |
> | -------- | ---- | ----------------- | ---------------- |
> | wikitext | 5.24 | 5.37 (+0.13)      | 5.29 (+0.05)     |
> | arxiv    | 3.78 | 3.99 (+0.21)      | 3.83 (+0.05)     |
>
> **Q3: Though SGD convergence proof is in the order, numerically it is usually more complicated in practice when diverse regularizers and different optimizers were used. Is there any commonly used regularization or optimizers that do not agree with AC-SGD?**
>
> **Response:** We added a paragraph at the end of Section 3.2 where we mention that, under some further Lipschitz assumptions on $b$ and the gradient on $b$, one can prove that if $f$ satisfies Assumptions A1 and A2, then $\tilde{f}(x) = f(x) + \frac{\lambda}{2}\|x\|^2$ satisfies Assumptions A1 and A2 with slightly different constants.
> In practice, we used $L_2$ regularization (thus weight decay) and Adam optimizer by default. Our method works well and we do not see obvious candidates for regularizers/optimizers on which it would not agree. We now mention this in Section 4.1.

---

### Official Review · Reviewer_6E4o · 2022-07-12

**Rating:** 5
**Confidence:** 2
**Soundness:** 2 fair
**Presentation:** 2 fair
**Contribution:** 2 fair

**Summary:**

This paper proposes an activation quantization method, AC-SGD, for the fine-tuning step on language models with slow network set-up. AC-SGD compresses the change of the activations instead of activation values directly. This paper shows this compression method can converge well and it can lead to throughput improvement under slow network systems.

**Questions:**

- How about model (tensor) parallelism (ex. Megatron)?


**Ethics Review Area:**

["I don’t know"]

**Limitations:**

- As I mentioned, this paper should describe why they assume this slow network system in detail. As far as I know, it seems to be not common.
- The experimental results are not sufficient to show that this method works well.
- Showing profiling results of the fine-tuning process with this method and various speeds of the network would be better for understanding.


**Strengths And Weaknesses:**

### Strengths
- This paper doesn’t aim at compressing activations directly. Instead, by compressing the changes of the activation, the proposed algorithm can achieve efficient convergence.
- With a slow network system (i.e., the portion of communication overheads is quite big during end-to-end distributed fine-tuning process), this algorithm can achieve higher throughput with smaller communication overhead.


### Weaknesses
As weaknesses of this paper, I have two concerns as below:
- I’m very confused about what slow networks are. It seems to be hard for me to imagine which scenario this paper assumes, V100 multi-gpus with 100Mbps network.  The only network I can imagine is that single GPUs are distributed and inter-connected with slow ethernet systems. It is so weird. I don’t understand why we should assume extremely slow interconnection between high-end GPUs for fine-tuning large PLMs.
- For the GPT results in Figure 4, I think showing training loss is not sufficient to prove that the compression method works well for generative language models, especially for fine-tuning steps. Since the pre-trained model consists of an enormous number of parameters and is trained with a large dataset, it is hard to trust the train loss and validation loss for fine-tuning dataset. It tends to be easily overfitted and there is rare correlation between loss results and measured scores. Since there is a limitation on scale of BERT-like models, this paper should show the inference results on generation tasks and large generation models.

---

> ### Author Response · Authors · 2022-08-02
> **Response to Reviewer (3/3)**
>
> **Q3: How about tensor parallelism (megatron)?**
>
> **Response:** We thank the reviewer for raising this interesting question. We have added a detailed discussion in Appendix F.
> In short: In tensor parallelism, activations are computed across different machines, and are then aggregated. Therefore we need to compress activations both before and after allreduce to support tensor parallelism. Specifically, suppose N machines conduct tensor parallelism, the output activation $A = A_1+A_2+…+A_N$ and we need to compress communication *twice* $A_Q = Q[ Q(A_1)+Q(A_2)+…+Q(A_N) ]$. We believe that delta compensation could be applied to all Q(-), similar to how previous work handles gradient compression (e.g. Eq. 3 and 4 in [1]). However, this requires careful further studies, both empirically and theoretically. We are very excited to explore this for activation compression in the future work.
>
> [1] Tang, Hanlin, et al. "Doublesqueeze: Parallel stochastic gradient descent with double-pass error-compensated compression." International Conference on Machine Learning. PMLR, 2019.
>
>
>
> **Q4:** **Showing profiling results of the fine-tuning process with this method and various speeds of the network would be better for understanding.**
>
> **Reponse:** Thank you for your suggestion! We here show the breakdown of AQ-SGD (fw4 bw8) per micro batch for GPT2-1.5B:
>
> | Bandwidth | Fw comp. | Bw comp. | Fw comm. | Bw comm. |
> |-----|-----|-----|-----|-----|
> | 500 Mbps  | 45ms     | 135ms    | 13ms     | 25ms     |
> | 200 Mbps  | 45ms     | 135ms    | 31ms     | 63ms     |
> | 100 Mbps  | 45ms     | 135ms    | 63ms     | 125ms    |
>
> We note that computation and communication can overlap, so the actual time is the larger one of the two. And we show that our algorithm can hide the communication in the GPU computation in most cases; only at 100Mbps bandwidth, the forward communication time is slightly longer than GPU computation.

---

> ### Author Response · Authors · 2022-08-02
> **Response to Reviewer (2/3)**
>
> **Q2:** **For the GPT results in Figure 4, I think showing training loss is not sufficient to prove that the compression method works well for generative language models, especially for fine-tuning steps. Since the pre-trained model consists of an enormous number of parameters and is trained with a large dataset, it is hard to trust the train loss and validation loss for fine-tuning dataset. It tends to be easily overfitted and there is rare correlation between loss results and measured scores. Since there is a limitation on scale of BERT-like models, this paper should show the inference results on generation tasks and large generation models.**
>
> **Response:** We agree with the reviewer that a better metric should be added to make our results stronger. We added (1) test perplexities and (2) a qualitative case study of the generated text, in Appendix I. Brief results are presented below, whereas full account is given in Appendix I :
>
> (1) First, we show the test perplexities after fine-tuning GPT2-1.5B. We can see that AQ-SGD achieves a similar test score as FP32, while DirectQ is significantly worse.
>
>
> |Text Perplexity| Wiki | Arxiv |
> |-----|-----|-----|
> | FP32 | 13.0 | 15.5 |
> | DirectQ fw4 bw8 | 13.7 (+0.7) | 15.9 (+0.3) |
> | AQ-SGD fw4 bw8 | 13.0 (+0.0) | 15.6 (+0.1) |
>
>
> (2) Furthermore, we also add a case study to better understand the generation quality across different methods. All methods are fine-tuned on WikiText with the same seed, and use greedy decoding to generate texts. We randomly pick documents from the test set of WikiText to construct the prompts and let the model complete the rest.
>
> **Prompt Case 1:** ... Matsuo Bashō, the greatest haiku poet, was also strongly influenced by Du Fu; … It is said that when he died
>
> - FP32: , his body was placed in a coffin with a Du Fu-like inscription.
> - AQ-SGD fw4 bw8: , his body was placed in a coffin with a Du Fu-like inscription.
> - DirectQ fw4 bw8: , his body was covered with a red cloth and his head was shaved.
>
> **Prompt Case 2:** ... Peppiatt notes that the exhibition showed Bacon no longer needed sensationalist material to make an impact, and was now
>
> - FP32: able to "make a serious contribution to the art of painting"
> - AQ-SGD fw4 bw8: able to "make a serious contribution to the art of painting"
> - DirectQ fw4 bw8: able to "make a serious statement about the state of the art"
>
> **Prompt Case 3:** ... At this point, Dokhturov's column appeared behind the French line and joined the battle. The French were outnumbered more than three to
>
> - FP32: one, but they held their ground and forced the Russians to retreat.
> - AQ-SGD fw4 bw8: one, but they were able to hold their ground.
> - DirectQ fw4 bw8: one, and the Russians were able to hold their ground.
>
> **Appendix I contains 10 more prompt cases.**
>
> In Cases 1 and 2, AQ-SGD produces the same results as FP32. In Case 3, although AQ-SGD does not generate the exact same output as FP32, it still expresses a similar meaning. This indicates our proposed compression scheme has relatively small impact on the learning process. In contrast, DirectQ may generate text that is not relevant to the context. For example, in Case 1, given the context ``Matsuo Bashō was strongly influenced by Du Fu'', we should expect the completed text to show the fact how he was influenced, just like FP32 and AQ-SGD did. But DirectQ fails to generate reasonable text based on the context.

---

> ### Author Response · Authors · 2022-08-02
> **Response to Reviewer (1/3)**
>
> We appreciate your great feedback! We have carefully thought through your questions and added corresponding experiments and detailed discussions in the updated paper. We provide details below:
>
> **Q1: I’m very confused about what slow networks are. It seems to be hard for me to imagine which scenario this paper assumes, V100 multi-gpus with 100Mbps network. The only network I can imagine is that single GPUs are distributed and inter-connected with slow ethernet systems. It is so weird. I don’t understand why we should assume extremely slow interconnection between high-end GPUs for fine-tuning large PLMs.**
>
> **Response:**  We have added more motivation and details about slow network setting (our target application scenarios) in Introduction and Appendix E.
>
> Our target application is decentralized training for large foundation models, a scenario that recently attracted intensive interests. Example projects include Learning@home [6], DeDLOC [1], and Training Transformers Together [2]. The goal of these projects is to enable a decentralized open-volunteering paradigm for foundation model training. As many geo-distributed users contribute their GPUs, these GPUs are connected via slow networks. For example, [1] investigates a heterogeneous with bandwidths of 200 Mb/s, 100 Mb/s, and 50 Mb/s; [2] advocates to train Transformer over the Internet with 10-100 Mb/s bandwidth; [3] considers network bandwidth less than 400Mb/s.
>
> In this setting, communication compression is key to performance. However, when compressing activations, existing methods rely on direct quantization. This inspired our paper, which provides the first activation compression method with rigorous theoretical guarantee and outperforms direct quantization.
>
>
> [1] Diskin, Michael, et al. "Distributed deep learning in open collaborations." Advances in Neural Information Processing Systems 34 (2021): 7879-7897.
>
> [2] Borzunov, Alexander, et al. "Training Transformers Together." NeurIPS 2021 Competitions and Demonstrations Track. PMLR, 2022.
>
> [3] Ryabinin, M., Dettmers, T., Diskin, M., & Borzunov, A. (2021). SWARM Parallelism: Training Large Models Can Be Surprisingly Communication-Efficient.
>
> [4] Yuan, Binhang, et al. "Decentralized Training of Foundation Models in Heterogeneous Environments." arXiv preprint arXiv:2206.01288 (2022).
>
> [5] ​​Beberg, Adam L., et al. "Folding@ home: Lessons from eight years of volunteer distributed computing." 2009 IEEE International Symposium on Parallel & Distributed Processing. IEEE, 2009.​​
>
> [6] Ryabinin, Max, and Anton Gusev. "Towards crowdsourced training of large neural networks using decentralized mixture-of-experts." Advances in Neural Information Processing Systems 33 (2020): 3659-3672.
>
> [7] Gpu economics cost analysis. https://venturebeat.com/2018/02/25/the-real-cost-of-mining-ethereum/.

---

> > ### Comment · Reviewer_6E4o · 2022-08-09
> > **Thank you for the responses and revisions**
> >
> >  Thank you for the detailed responses and revisions. As I respect the authors' efforts and other reviewers' opinions, I will raise my point to 5. But, I still have a concern on the assumption (i.e., distributed computing system with very slow network + fine-tuning LM). Since PLM is big enough to represent the fine-tuning dataset, compressing or low-precision training during the fine-tuning process is relatively easier than other conventional training processes. I think this paper's novelty depends on how realistic this assumption is. For this aspect, I will defer to AC or other reviewers' thinking.

---

### Author Response · Authors · 2022-08-02
**Revision Summary**


We thank all the reviewers for their comments and suggestions in helping improve the quality of the paper. We were glad that reviewers found the algorithm **interesting, effective and novel** (R3), the theory with **proven convergence** (R1, R2, R3, R4), the presentation **clear** (R2), and the experiment **extensive and evaluated against strong baselines** (R3, R4). We have updated the paper (changes in blue) to incorporate these insightful feedbacks. We summarize the major changes as follows:

1. [R1,3] better motivation of slow networks in Section 1 and Appendix E;
2. [R2,3,4] better discussion on limitation and future work in Appendix G;
3. [R1] a new case study of generation results in Appendix I;
4. [R2] a new experiment on training from scratch in Appendix H.3;
5. [R2] a new experiment on FP16 training in Appendix H.4;
6. [R2] a discussion on theory with regularization in Section 3.2;
7. [R1] a discussion on tensor parallelism in Appendix F;
8. [R2] changed “activation compression” to “activation quantization”, and “AC-SGD” to “AQ-SGD”.

We look forward to the discussion phase with all the reviewers. We appreciate all reviewers’ comments and feedback!

---

### Meta-Review · Area_Chair_p1zi · 2022-08-26

**Recommendation:** Accept
**Confidence:** Less certain

**Metareview:**

In this paper, authors propose to speed up the fine tuning of large models over slow networks by compressing *deltas* of activations (vs activations themselves), so as to reduce the computation cost.

Original reviews were mixed, but at the end of the discussion period, all reviewers are leaning towards acceptance. The main issues that were raised are:
* The motivation for training very large models over slow networks
* The limited amount of metrics to validate the quality and robustness of the optimization process
* Concerns about the scalability of the method (storage requirements) and its applicability to the online setting

I consider that these concerns have been mostly addressed during the discussion period by the authors, who also remained honest about some of the limitations of their method.

In my opinion, the pros of this work (a practically useful idea) outweigh the cons (it may only be useful in somewhat niche settings), and I thus recommend acceptance.

**Award:**

No

---

### Decision · Program_Chairs · 2022-09-14

Accept